# Digital circuits and neural networks based on acid-base chemistry implemented by robotic fluid handling

Ahmed A. Agiza [1] ✉, Kady Oakley[1], Jacob K. Rosenstein [1],
Brenda M. Rubenstein [1], Eunsuk Kim[1], Marc Riedel[2] & Sherief Reda[1]

Acid-base reactions are ubiquitous, easy to prepare, and execute without sophisticated equipment. Acids and bases are also inherently complementary and naturally map to a universal representation of "0" and "1." Here, we propose how to leverage acids, bases, and their reactions to encode binary information and perform information processing based upon the majority and negation operations. These operations form a functionally complete set that we use to implement more complex computations such as digital circuits and neural networks. We present the building blocks needed to build complete digital circuits using acids and bases for dual-rail encoding data values as complementary pairs, including a set of primitive logic functions that are widely applicable to molecular computation. We demonstrate how to implement neural network classifiers and some classes of digital circuits with acid-base reactions orchestrated by a robotic fluid handling device. We validate the neural network experimentally on a number of images with different formats, resulting in a perfect match to the *in-silico* classifier. Additionally, the simulation of our acid-base classifier matches the results of the *in-silico* classifier with approximately 99% similarity.

Semiconductor devices have led the information technology revolution over the past several decades. However, there are good reasons to explore alternative methodologies for information storage and data processing, such as their potential for greater power efficiency, greater affordability, biological compatibility, and ability to adapt to different environmental conditions that may be inhospitable to conventional semiconductor technologies. One interesting research direction is a molecular computing paradigm[1,2] that relies on chemical reactions, which can be considerably more space- and energy-efficient than digital approaches[3].

Various methodologies have been proposed for molecular computation. For example, some demonstrations have leveraged chemical reaction-diffusion processes, often using the Belousov-Zhabotinsky (BZ) reaction[4] as a chemical oscillator[5]. While these oscillators' dynamics can be used to perform tasks such as classification[6], they incorporate complex spatio-temporal signals rather than stable endpoints, and they are sensitive to factors such as active sample mixing and temperature. Other hybrid approaches utilize electronic actuators to assist in encoding the inputs for reaction initiation[6]. A different approach is proposed by Arcadia et al. where they model the autocatalytic reactions as an activation function used in artificial neural networks[7]. They demonstrate an autocatalysis-based digital image recognition model with the images encoded in the concentration of a catalyst. In this work, we use acid-base concentration to encode digital information.

DNA computing has been the most widely studied form of molecular computing[2,8–10]. It relies on DNA strands to carry information and execute different logic and arithmetic operations. Sequence-specific hybridization among networks of DNA sequences is used to model the algorithm chosen to solve the given problem. Nevertheless, DNA has limitations such as temperature sensitivity, long computation times, and the need for large numbers of custom

[1]Brown University, Providence, RI, USA. [2]University of Minnesota, Minneapolis, MN, USA. ✉e-mail: ahmed_agiza@brown.edu

reagents and optimizations to perform experimental demonstrations of computation.

In this work, we use a chemical approach that employs acid-base reactions as a means for universal computations orchestrated by an acoustic liquid handler.

The proposed chemical system is straightforward. A mixture of a strong acid (HX) and a strong base (YOH) can be summarized as reaching an equilibrium between three reactions:

$$HX + H_2O \rightarrow H_3O^+ + X^- \tag{1}$$

$$YOH \rightarrow Y^+ + OH^- \tag{2}$$

$$H_3O^+ + OH^- \rightleftarrows 2H_2O \tag{3}$$

An interesting property of this simple chemical system is provided by the neutralization reaction (Eq. (3)), which couples the results of the acid and base reactions, primarily leaving behind only the more abundant species. Let $x_i$ denote the type of droplet $i$ (e.g., acid or base). If we consider a mixture of an odd number of $n$ droplets with equal volume and concentrations, with $m$ of these droplets being acids, then:

$$Majority(x_1, x_2, \ldots, x_n) = \begin{cases} Acid\ (= +1) & \text{if } 2m > n \\ Base\ (= -1) & \text{if } n > 2m, \end{cases} \tag{4}$$

which is analogous to the majority function in Boolean logic.

Importantly, the majority function combined with the inversion operation form a functionally complete set[11], which means these operations are sufficient to model any Boolean logic function.

However, the inversion operator is challenging to realize in chemical computation since inversion would require a conditional bistable chemical network which can be toggled without knowledge of the current state. As best we know, there is no simple reaction that would both change an acid to a base, and change a base to an acid.

To obviate the need for complex reaction networks, we can take advantage of the symmetry of Equations (1) and (2), and recognize an opportunity to encode information in the difference between acidic and basic solutions.

Thus, we propose a dual-rail encoding that represents data values as complementary pairs, e.g., (acid, base) or (base, acid). An encoded value will be represented by a concentration of $[H_3O^+] = x$ ions ($pH = -\log x$) in one rail and a concentration of $[OH^-] = x$ ions ($pH = 14.0 - \log x$) in the complementary rail. Differential encoding supports the inversion operation by simply exchanging the roles of the complementary rails during computations. Computations performed in the dual-rail form conveniently produce both the output and its logical complement, similar to some common digital structures like the D Flip-Flop[12]. Figure 1 illustrates some of the fundamental building blocks of our methodology.

The concept of complementary representations, such as dual-rail encoding, has been used to solve relevant problems in other computational systems[13,14]. For example, Jiang et al. used complementary solutions with different concentrations to represent bits within their approach[13]. Similarly, Qian et al. used the notion of complementary inputs and gates to design seesaw gates[14]. Our approach extends the dual-rail information concept to universally relevant acid/base chemistry.

## Results
### Encoding data in a dual-rail, acid-base representation

As illustrated in Fig. 1, our encoding approach represents each bit using a pair of complementary solutions where the pair (Acid, Base) represents the value "1" or TRUE, while the pair (Base, Acid) represents the

value "−1" or FALSE. A pair with neutral pH 7 represents the midpoint of the full scale, which is the boundary between TRUE and FALSE, or the value "0" when applicable. In some applications, we can further extend the encoding to a larger set of discrete values by controlling the concentration of the encoded values. For example, to build a 3-bit encoding system with eight possible non-zero values, we can discretize the data symmetrically into four negative values {−4, −3, −2, −1} and four positive values {1, 2, 3, 4}. The positive values would be encoded as (Acid, Base), and the negative values would be encoded as (Base, Acid).

For each data value, we dilute the solution by adding an amount of water that is equal to:

$$Added\ water\ volume = \left( initial\ volume \times \frac{maximum\ encoded\ value}{encoded\ value} \right) - initial\ volume \tag{5}$$

We make these solutions experimentally by providing an acoustic liquid handler (Echo 550, Beckman Coulter) with stocks of acids, bases, and deionized water. Each computational operation involves transporting portions of the input solutions to an output well plate. The role of the liquid handler is analogous to the wires in an electronic circuit. As shown in Fig. 1c, for each value in the information sequence being encoded, if the value is positive, we instruct the liquid handler to dispense an acid droplet followed by a base droplet in the next available slots in the destination grid. Similarly, if the value is negative, the liquid handler dispenses a base droplet followed by an acid droplet, instead. Finally, if the computation involves more discretized values, the liquid handler dispenses water droplets into the corresponding wells to dilute the solutions as described in equation (5). Finally, we add a pH indicator to the final outputs of the computation to read the results. If the indicator measures an (Acid, Base) pair, this signifies that the output is TRUE, while a (Base, Acid) pair would indicate FALSE.

**Modeling digital circuits using acid-base reactions.** As discussed in the Introduction, when mixed, acids and bases naturally execute the majority function and inversion can be realized by exchanging the roles of the complementary pair. Since the majority and inversion operations are functionally complete[11], they can be used to represent any logic function. Figure 2 depicts primitive AND, OR, INV, NAND, and NOR logic gates represented in terms of acid-base operations. In order to avoid neutral pH outputs, all of the building blocks should have an odd number of inputs. To construct two-input logic gates, the third input is a constant bias. Table 1 shows the truth table for the AND gate and the corresponding representation using acid-base reactions with a dual-rail encoding.

Figure 3 depicts a complete logic circuit for the digital decoder using acid-base blocks. Figure 3a shows the original logic circuit, while Fig. 3b shows the corresponding circuit using the equivalent acid-base logic blocks. Finally, Fig. 4 depicts the implementation of the decoder circuit using the liquid handler. Figure 4a shows the truth table for the 2-bit decoder circuit, Fig. 4b shows the first level of the circuit that includes the original input and its complements in addition to acid and base stocks to be used as biases for the AND gates, and Fig. 4c shows the final outputs of the circuits after applying the AND gate according to the circuit design.

One limitation of these acid-base logic gates is that the outputs of successive cascaded logic stages have progressively more neutral pH because of the neutralization reaction. Conceptually, one way to resolve this is to simply replace the intermediate results with fresh acid or base solutions. For simplicity this is what we assume in logical simulations; however, it implies interrupting the computation. Other future alternatives to restore the pH logic levels could include incorporating mechanical valves with pH-sensitive materials, which we highlight in the Discussion.

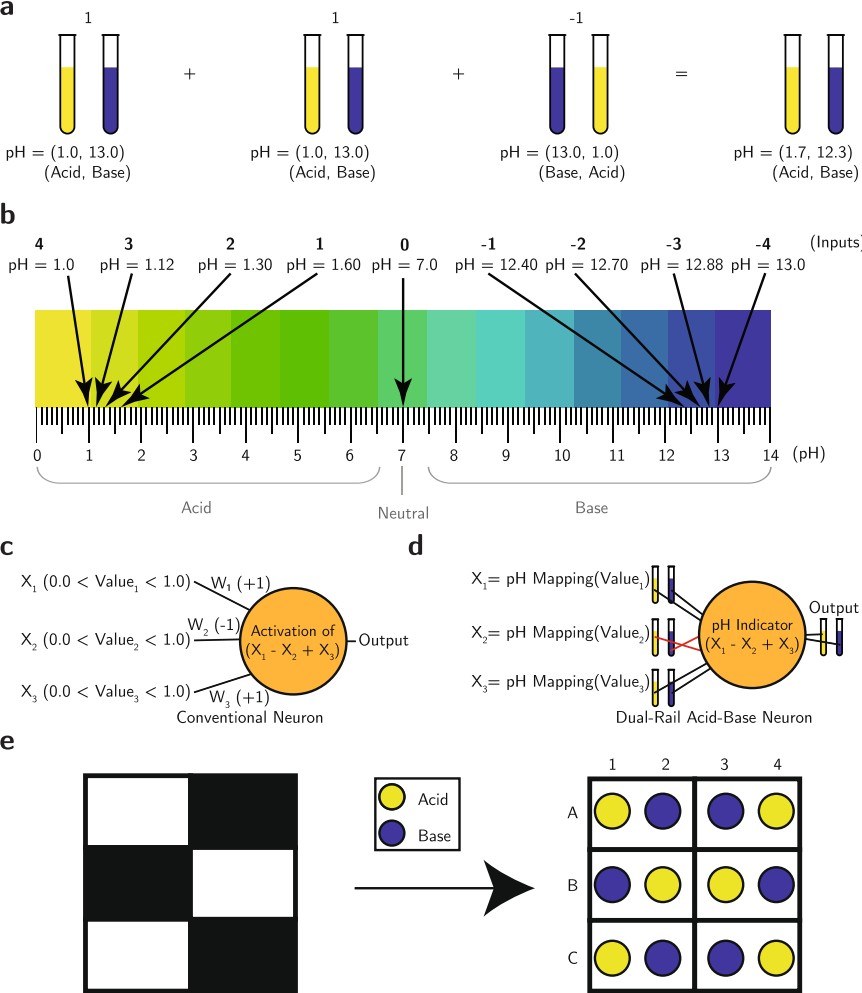

Fig. 1 | Method overview. a The dual-rail acid-base encoding for an addition followed by a subtraction (using the inverted representation). Each "1" is represented by an acidic solution (on the pH rail) and a basic solution (on the complementary pH rail); a "−1" is represented by the same solutions in reverse order. b Illustration of the mapping of discretized inputs into their corresponding pH values.

c Conventional neuron with its default operations. d Equivalent neuron using an acid-base encoding with analog input values mapped to the corresponding pH value from the calculated range. Crossing lines indicate multiplying by −1.
e Encoding of a 2 × 3 binary array of pixels into a well plate of acids (yellow) and bases (blue), where each pixel value is mapped to two wells.

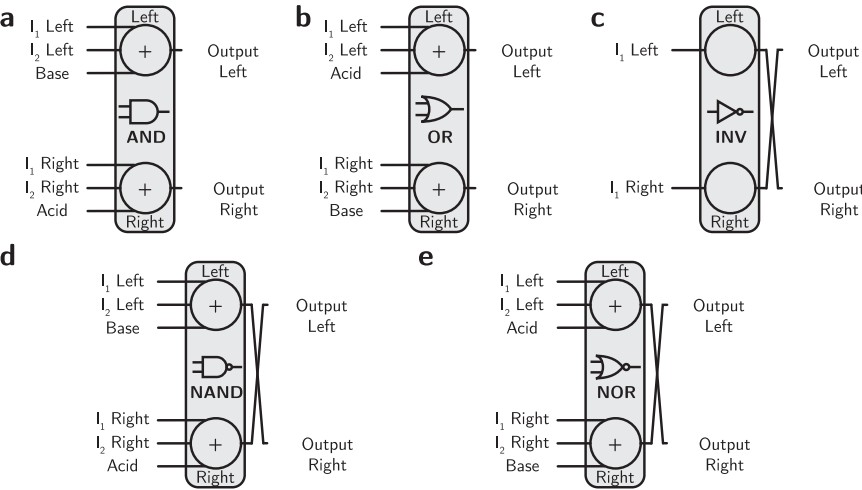

Fig. 2 | Primitive logic gates represented by acid-base blocks. a AND gate, b OR gate, c INV gate, d NAND gate, e NOR gate. Each gate with an even number of inputs requires a third biasing value to implement its function and eliminate neutrality.

The + sign indicates that the input solutions are mixed. The crossing output lines indicate that an inversion has been performed.

**Table 1 | The truth table for the AND gate in digital systems and its corresponding representation in our dual-rail acid-base method**

| Digital AND Gate | | | Dual-rail acid-base operation | | | |
|---|---|---|---|---|---|---|
| In 1 | In 2 | Out | Input 1 | Input 2 | Bias | Output |
| 0 | 0 | 0 | Base/Acid | Base/Acid | Base/Acid | Base/Acid |
| 0 | 1 | 0 | Base/Acid | Acid/Base | Base/Acid | Base/Acid |
| 1 | 0 | 0 | Acid/Base | Base/Acid | Base/Acid | Base/Acid |
| 1 | 1 | 1 | Acid/Base | Acid/Base | Base/Acid | Acid/Base |

The first two columns correspond to the input permutations. The following fourth and fifth columns represent the acid-base encoding. The sixth column represents the constant bias input used with the AND gate. The last columns on either side show the result (output) of mixing the inputs in the acid-base AND gate.

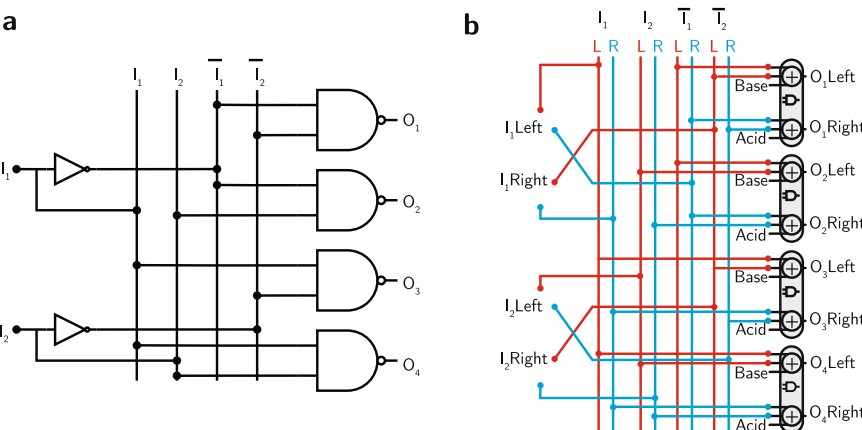

**Fig. 3 | Digital decoder and its equivalent acid-base implementation. a** The decoder circuit using digital logic blocks. **b** The equivalent decoder circuit using acid-base blocks.

**MNIST digit classifier using acid-base reactions.** Neural networks are computing systems that derive relations from available datasets through weighted sums of input features executed through connected neurons. Classification is a common task for neural networks to classify inputs (e.g., hand-written digit image) into a set of classes (e.g., the corresponding digit number). Additionally, the network is customized for different conditions, such as (1) The input image format: colored, grayscale, or black and white. We ran our experiment on black and white, and 3-bit grayscale images. (2) The type of training weights. We chose binary weights, which are limited to "1" and "−1," since we can readily represent a "1" as a direct transfer of an input and a "−1" as an inverted transfer that flips our rails in our encoding. We used softmax[15] as an activation function for all our models. Finally, the neural network is split into two phases: a training phase and an inference phase. The training phase is a computationally-intensive phase that involves using a prepared dataset to find optimal weights that generate the desired labels with high accuracy. The inference phase uses the pre-trained weights to predict labels for new inputs outside of the dataset. Much like previous DNA computing demonstrations[9,16], our work focuses on using acid-base chemistry to realize the inference phase based on pre-trained weights, which is the main phase in most practical applications. The acid-base encoding presented here computes the weighted sum performed by each neuron by mixing strong acids and bases that represent the encoded image based on the binary weights of the network. A black pixel is represented by a (Base, Acid) pair, while a white pixel is represented by an (Acid, Base) pair. Our approach performs the multiplication by "1" (through a direct liquid transfer without inversion) or by "−1" (through swapping the order of the encoded dual-pair value during the transfer). The resulting solution is acidic or basic, which maps to the classification result of the network where (Acid, Base) pairs represent positive outputs, while (Base, Acid) pairs

represent negative outputs. Additionally, the pH indicator serves as a nonlinear activation layer that computes the neuron's output based on the pH threshold.

We trained a binary neural network consisting of two fully-connected neurons with the softmax activation function to recognize images that represent hand-written digits. In our experiments, we choose images corresponding to the hand-written 0 and 1 digits from the MNIST (Modified National Institute of Standards and Technology) digit classification dataset[17]. The pixels of the input image are flattened sequentially (left to right, and top to bottom) into a 1D vector. A copy of the 1D vector representation of the image is fed into each neuron. Figure 5 shows the 2-neuron network for the classification of $8 \times 8$ images into zero or one. After training the weights, we perform our classification as shown in Fig. 6. Figure 6a shows an example image from the validation dataset. We start by encoding the image into its acid-base representation shown in Fig. 6b using the acid-base stock. We encode the image as many times as the number of neurons (classes) in our approach (we have two classes/neurons since we are classifying the digits "0" and "1"). After encoding and applying the weights, we compute the output from each neuron to decide whether it is a "1" or "−1." Finally, as shown in Fig. 6c, for each neuron, we pool (sum) all of the pH rails of the dual-rail to compute the pH rail of the neuron's output, and all of the complementary pH rails of the dual-rail to compute the complementary pH rail of the neuron's outputs. We add a pH indicator to the neuron outputs to determine whether the outputs are acidic or basic based on their color. A neuron with an output pair (Acid, Base) indicates that the input image matches the digit this neuron is supposed to recognize. For instance, the first neuron is supposed to recognize the digit 0, while the second neuron is supposed to recognize the digit 1. In contrast, a neuron with an output pair

**a**

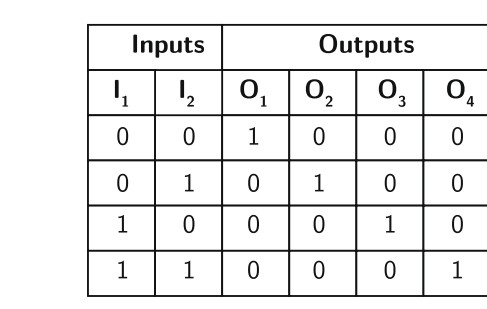

| Inputs | | Outputs | | | |
|:---:|:---:|:---:|:---:|:---:|:---:|
| $I_1$ | $I_2$ | $O_1$ | $O_2$ | $O_3$ | $O_4$ |
| 0 | 0 | 1 | 0 | 0 | 0 |
| 0 | 1 | 0 | 1 | 0 | 0 |
| 1 | 0 | 0 | 0 | 1 | 0 |
| 1 | 1 | 0 | 0 | 0 | 1 |

**b**

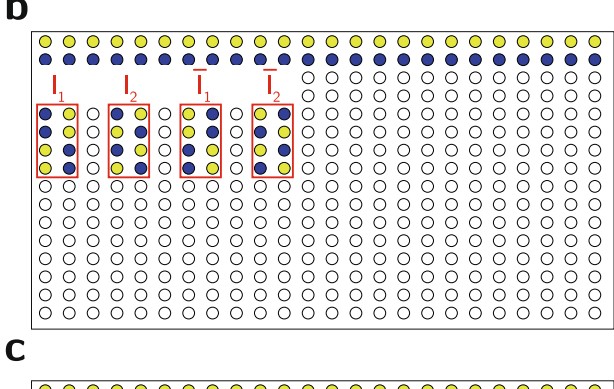

**c**

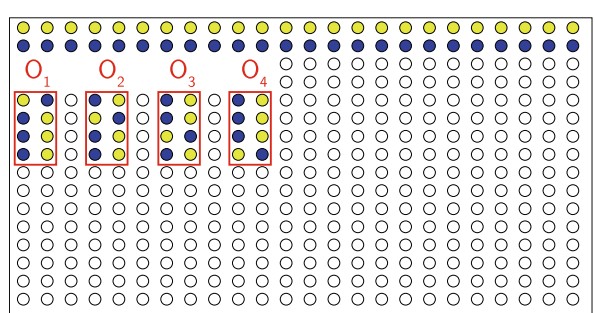

**Fig. 4 | Implementation of the 2-Bit Decoder using the liquid handler. a** The truth table for the 2-bit decoder. **b** The initial acid (yellow) and base (blue) stocks of the encoded inputs and their complements. **c** The final output of the acid-base circuit after applying the AND gate.

**a**

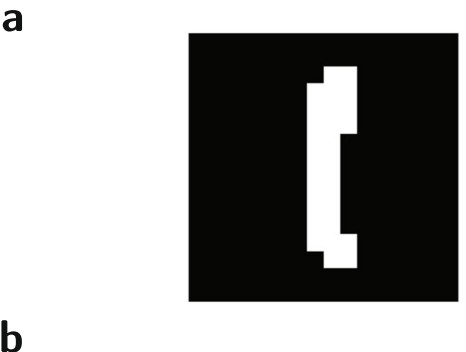

**b**

Encoded image after applying the first neuron

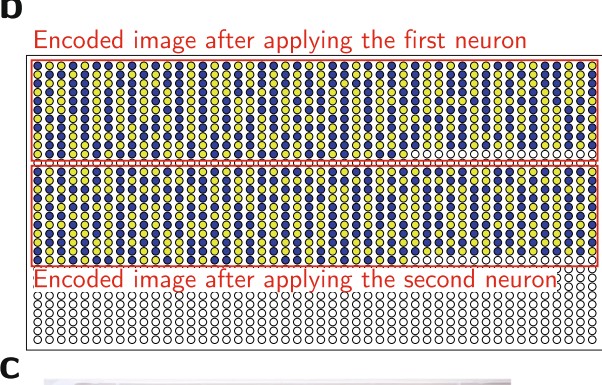

Encoded image after applying the second neuron

**c**

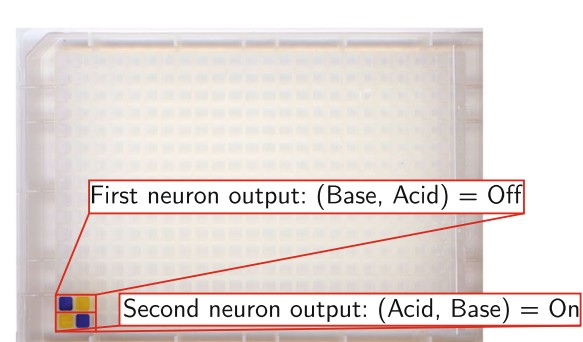

First neuron output: (Base, Acid) = Off

Second neuron output: (Acid, Base) = On

**Fig. 6 | Classification of the digit "1" using an acid-base network. a** The original 16 × 16 image to be classified. **b** The encoding of the image and inversions using acids (yellow) and bases (blue). **c** The final classification output image from the lab experiment.

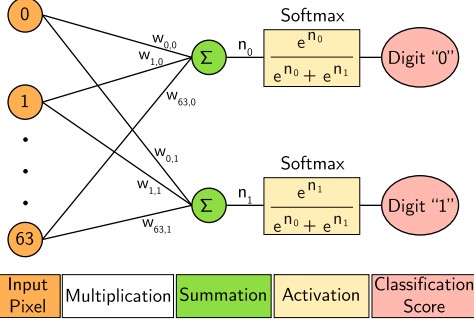

**Fig. 5 | Network architecture.** Neural network for the classification of 8 × 8 images into the digits zero or one.

of (Base, Acid) indicates that the input image does not match the digit this neuron is supposed to recognize.

We ran our model in simulation using various parameters (image size, discretization levels, and the number of classes) across the full validation set and calculated the match percentage between our

experimental reactions and our theoretical pH simulations. Additionally, we picked ten random samples from the validation dataset to evaluate the model experimentally using the liquid handler. The minor reduction in accuracy from the acid-base model compared to the in silico model arises from relying on the pH color indicator for reading the outputs. We rely on the color to determine the state of each neuron where (Acid, Base) = +1 and (Base, Acid) = −1; which is sufficient when the in silico model produces scores that have a positive sign for the correct label and a negative sign for the wrong label. However, for a few samples, the in silico model and the acid-base model will generate matching results, except that all outputs are either (Acid, Base) or (Base, Acid). However, when we measure the actual pH levels, the output with the highest concentration always matches the prediction label from the in silico model. Supplementary Figure 1 shows an 8 × 8 image example that cannot be classified correctly with the acid-base network; the pixel values for the image are shown in Supplementary Table 1.

For the first set of results, we ran our experiment using binary pixel values and two classes using dataset images of sizes 8 × 8, 12 × 12, 16 × 16, and 28 × 28. As shown in Table 2, our chemical-based classifier is able to match theoretical results from the digital network with 99.61% similarity on average across different input sizes. The model's

**Table 2 | Matching accuracy for 2-class, binary image classifier**

| Image size | Model accuracy (%) | Acid-base network match (%) |
|---|---|---|
| 8 × 8 | 95.45 | 99.81 |
| 12 × 12 | 95.59 | 99.39 |
| 16 × 16 | 95.82 | 99.39 |
| 28 × 28 | 96.00 | 99.86 |
| Average | 95.72 | 99.61 |

Image size, classification accuracy of the digital model, and matching accuracy between the digital model and the theoretical pH values. Source data are provided as a Source Data file.

**Table 3 | Matching accuracy for 2-class, 3-bit image classifier**

| Image size | Model accuracy (%) | Acid-base network match (%) |
|---|---|---|
| 8 × 8 | 93.28 | 97.26 |
| 12 × 12 | 95.15 | 99.15 |
| 16 × 16 | 96.28 | 99.48 |
| 28 × 28 | 96.37 | 99.86 |
| Average | 95.27 | 98.94 |

Image size, classification accuracy of the digital model, and matching accuracy between the digital model and the theoretical pH values. Source data are provided as a Source Data file.

**Table 4 | Matching accuracy for 3-class, binary image classifier**

| Image size | Model accuracy (%) | Acid-base network match (%) |
|---|---|---|
| 8 × 8 | 95.68 | 96.12 |
| 12 × 12 | 95.86 | 98.82 |
| 16 × 16 | 96.00 | 98.82 |
| 28 × 28 | 96.05 | 99.62 |
| Average | 95.90 | 98.35 |

Image size, classification accuracy of the digital model, and matching accuracy between the digital model and the theoretical pH values. Source data are provided as a Source Data file.

accuracy is calculated over the validation dataset, and it is defined as the ratio of number of correct classifications to the number of samples in the validation dataset. The match similarity is calculated over the validation dataset using pH simulation, and it is defined as the ratio of the number of of samples where the in silico and acid-base networks match to the number of samples in the validation dataset.

For the second evaluation, we extended our experiment to classify 3-bit images (8 grayscale levels) instead of binary values. We use a network that is similar to the previous experiment but uses 3-bit values instead of binary ones. We discretized the input pixel values into eight different integers where each pixel value $x$ is mapped to the integer: $\text{Round}(8 \times \frac{x}{255.0}) - 4$. We then utilized the dilution equation explained before to represent the given integer. We re-trained the network with the updated representation. The remaining flow matches the previous experiment. As shown in Table 3, our chemical-based classifier is able to match the theoretical results from the digital network with 98.94% similarity on average across different input sizes.

For the final set of results, we extended our experiment by using three fully connected neurons instead of two to classify binary images belonging to three different classes/digits ("0", "1", and "2"). As shown in Table 4, our chemical-based classifier is able to match the theoretical model's results from the digital network with 98.3% similarity on

average across different input sizes. Supplementary Tables 2–41 give the weight values for every model used in the evaluation.

## Discussion

In this work, we explored the usage of acid-base reactions as a basis for complex computation orchestrated by a robotic fluid handler to perform the encoded reactions. We introduced a dual-rail encoding method that suits the natural complementarity of acids and bases and enables our method to support the negation operation, which is essential for realizing universal computation. We leveraged the fact that the pH of a strong acid/base system is dominated by whichever species is more concentrated to realize the majority function in chemistry.

Our approach presents a system for modeling various computations without using the complex spatio-temporal encodings presented in other unconventional computing methods. Furthermore, the reactions enable rapid computation, given the speed of strong acid/base reactions, without expensive equipment or intricate handling to execute the operations. Straightforward computation using acid-base chemistry can provide new ways of interacting with molecular data storage systems, for example, by interacting with pH-dependent protection/deprotection groups that could be incorporated into DNA[9,18] or small-molecule information[19–21] storage systems.

As an illustration of our approach, we implemented an image classification neural network using a system of acid-base reactions. We utilized binary neural networks and binary cross-entropy loss functions to generate complementary outputs compatible with our method that can be distinguished using a pH indicator. As a result, we managed to classify the MNIST digits dataset (for two and three digits). Simulated experiments based on our approach classified images with a 99% accuracy. We also extended our encoding and computational approach to work with non-binary inputs. Using the dynamic range of possible concentrations, we can generate more interesting representations for our data to support more efficient computations[22].

Looking forward, we are investigating the possibilities of other nonlinear chemistries. This work presented only strong acids/bases, but incorporating weaker acids and bases as pH buffers[23] could introduce nonlinearity, thresholds, and plateaus in the pH curves. Introducing nonlinear pH operators could produce more complex primitives, such as activation functions[7], which are a critical part of multilayer neural networks. By developing a cascadable chemical nonlinear activation function, we would be able to extend our method to support deep neural networks with multiple hidden layers.

In addition to neural networks, we also presented an approach for using the acid-base reaction to represent digital logic circuits. We designed multiple digital logic primitives using the acid-base system while introducing a third biasing input to the blocks to guarantee a non-neutral output and the desired functionality.

Since acid-base reactions can be diluted when propagated across multiple layers or gates, we proposed solving dilution problems by manually replacing the intermediate results with fresh non-diluted stocks of acids or bases. However, this step can be automated by combining with other existing concepts, such as the usage of pH-sensitive hydrogel valves[24] to build a restoration mechanism to maintain the computation across a longer chain of reactions.

Additionally, digital logic optimization algorithms can be incorporated to improve the quality of our approach. For example, we can optimize the computations to reduce the number of logic layers in the digital circuit to minimize the dilution that results from cascading multiple reactions during the computation. Furthermore, our procedures can be incorporated into other binary systems of a similar nature. For example, the Hadamard transform[25] uses operations analogous to those employed in this work.

Our dual-rail representation provides a methodology to solve the challenges of representing negation in chemistry, that can be extended

beyond acid-base reactions. A complementary representation can serve as a useful alternative for any computational system that can not readily negate values. Furthermore, the relationship between the two (or more) rails does not have to be inverse but rather any desired relationship hard to realize in a single rail system. For example, the associated rails can represent such operations as multiplication by a scalar or a pre-computed trigonometric function.

## Methods

### Materials and reagents

Hydrochloric Acid (HCl, 36.5–38.0%, Fisher Chemical) and Sodium Hydroxide (NaOH, ≥97%, Fisher Chemical) were dissolved using deionized water as a solvent (Millipore Milli-Q) to a concentration of 100 mM. Bromothymol blue ($C_{27}H_{28}Br_2O_5S$, 0.04%, VWR Chemicals) was used as an indicator for the pH of the solutions. Deionized water was made available to the liquid handler for dilution of the reagents in the 3-bit image neural network classification experiments.

### pH simulator

In order to verify our results on the full dataset, an automated pH simulator was designed to simulate the reaction outcomes for the full dataset. The simulator predicts the pH by calculating the hydrogen ion concentration [$H^+$] that results from mixing the two or more aqueous solutions involved in our computations. The simulator was designed to directly accept the liquid handler's programming files to recreate the liquid handler's transfers and eliminate related human errors.

### Data preparation and network training

The MNIST dataset was used for training the acid-base digit classifier due to its well-understood properties and clear benchmarks. The dataset provides 70,000 grayscale hand-written digits ("0"–"9") of size 28 × 28 (784 total values, where each value is between 0 and 255). Data points for the digits "0", "1", and "2" were used for the network including 5923, 6742, and 5958 points for training, and 980, 1135, and 1032 points for validation, respectively. The whole dataset was used for training and simulation in silico, while a random sample from the validation dataset was used for the experimental verification as described in the next section. Each experiment was evaluated using four different image scales: 8 × 8, 12 × 12, 16 × 16, and 28 × 28. Two variants were provided for each image: the binary variant where each pixel was given a value of 1 or −1 based on a threshold of 128 where pixels of intensity less than 128 were assigned an encoding of (Base, Acid) while pixels of value greater than or equal to 128 were assigned an encoding of (Acid, Base), and the 3-bit variants where each pixel was discretized to a value from {−4, −3, −2, −1, 1, 2, 3, 4} representing the pixel intensity ranges {0:32, 33:64, 65:96, 97:128, 129:160, 161:192, 193:224, 225:256}, respectively. The same processed dataset image files were also used by the pH simulator to compute the accuracy of the acid-base experiments by computing the resultant pH after running each image through our acid-base neural network and comparing it to the expected classification. The dataset was split into training and validation sets according to an 85/15 ratio. The neural network model was trained with the weight binarization technique by Courbariaux et al.[26] to constraint the weights to "+1" and "−1", as explained above. A learning rate of 0.001 and a *sigmoid* activation function were used to train a neural network consisting of one classification layer for 30 epochs. In addition, the Binary Cross Entropy was used as a loss function. After training multiple networks for the different permutations of the image scale and data scale variants, the weights were exported into text files to be used to guide our acid-base experiments.

### Experimental verification of the acid-base neural networks

The experiments were conducted using the acoustic liquid handler. The handler was provided with an initial stock 384-well polypropylene

**Table 5 | The run-time in minutes for a single input of the three experiments: 2-class classification for binary images, 3-class classification for binary images, and 2-class classification for 3-bit images**

| Image size | 2-class binary image | | 2-class 3-bit image | | 3-class binary image | |
|---|---|---|---|---|---|---|
| | Encoding | Pooling | Encoding | Pooling | Encoding | Pooling |
| 8 × 8 | 13 min | 1 min | 13 min | 1 min | 19 min | 2 min |
| 12 × 12 | 29 min | 3 min | 29 min | 3 min | 44 min | 4 min |
| 16 × 16 | 52 min | 5 min | 52 min | 5 min | 78 min | 8 min |
| 28 × 28 | 158 min | 16 min | 158 min | 16 min | 237 min | 24 min |

plate (minimum volume 10 μL, maximum volume 60 μL) that is manually prepared with rows of acidic solutions (HCl) followed by rows of basic solutions (NaOH) and then by rows of deionized water. Each well was filled with 50 μL of the designated solution. The liquid handler has a maximum deviation of 10% from its target volume. For the first phase, using the image exported from the processed MNIST dataset, the liquid handler is programmed to encode the image using a series of transfers from the initial stock plate into a 1536-well low dead volume plate (minimum volume 1 μL, maximum volume 4 μL). The process is repeated according to the number of neurons in the neural network model. For each pixel in the image, if the value is negative and the neuron weight is +1 or the value is positive, and the neuron weight is −1, the handler transfers 2.4 μL of basic solution into the first empty well, followed by 2.4 μL of acidic solution into the next well. Similarly, if the value is negative and the neuron weight is −1 or the value is positive, and the neuron weight is +1, the handler transfers 2.4 μL of acidic solution into the first empty well, followed by 2.4 μL of basic solution into the next well. Afterward, if more values other than "1" and "−1" are supported, the handler dispenses quantities of the deionized water, as described in equation (5), into the corresponding wells in the destination to dilute the solutions according to their discretized pixel values.

The final phase is computing the output of each neuron by pooling the pH and complementary pH rails of each dual-rail pair. The summation is carried out from the encoded image in the 1536-well plate back into the 384-well plate. For each neuron, we transfer 200 nL from the pH rail of each pixel's dual-pair (starting from the first well, scanning to the end of the neuron wells, and skipping every second well) into the well assigned to the pH rail of the output of that neuron. We do the same for the well representing the complementary pH rail of each pixel's dual pair. The overall time to pool the neuron outputs was around 5 min. Table 5 gives the breakdown of the run-time of the two stages of the experiment: image encoding and result pooling. The shown times are for the three different experiments presented: 2-class classification of binary images, 3-class classification of binary images, and 2-class classification of 3-bit images. The run-time of the experiment depends on the number/volume of liquid transfers needed to execute the experiment. The 2-class binary Image takes the same amount of time as the 3-bit grayscale variant since the transferred volumes are equivalent (the 3-bit grayscale variant only substitutes some of the transferred acid/base with water for dilution). On the other hand, the 3-class variant has an extra run-time overhead since it requires more liquid transfers for the extra neuron (the 3-class network has three neurons, compared to two neurons for the 2-class network). Supplementary Figure 2 shows an illustration of encoding 3-bit grayscale images using the approach described earlier. The color of each well corresponds to the pH color shown in Fig. 1b. All the validation datasets with different scale permutations were verified by running through the pH simulator for theoretical validation (without noise or experimental errors). We selected ten random images from the validation dataset

for the real experimental evaluation by the Echo liquid handler as follows: one $8 \times 8$ binary image (for the 2-class network), one $8 \times 8$ binary image (for the 3-class network), one $12 \times 12$ binary image (for the 2-class network), one $12 \times 12$ binary image (for the 3-class network), and two $16 \times 16$ binary images (for the 2-class network) two $16 \times 16$ 3-bit images (for the 2-class network). Supplementary Figure 3 and Supplementary Movie 1 show a walkthrough of the experiment's setup. Additionally, we have performed an experiment of digital networks on a different platform described in Supplementary Note 1, Supplementary Fig. 4, and Supplementary Movie 2.

### Reading the classification values

The final phase of the experimental validation is reading out the neuron output. $5\,\mu L$ of the Bromothymol blue pH indicator were added to the dual-rail output of each neuron; a blue color indicated a basic solution, while a yellow color indicated an acidic solution. Each neuron's dual colors represented its final classification where a neuron's output of (Acid/Yellow, Base/Blue) indicated it is a "1" (the predicted classification) while (Base/Blue, Acid/Yellow) indicated it is a "−1."

### Reporting summary

Further information on research design is available in the Nature Portfolio Reporting Summary linked to this article.

## Data availability

The processed MNIST datasets are available at https://doi.org/10.6084/m9.figshare.21753545.v4[27]. Source data are provided with this paper.

## Code availability

The software is available at https://doi.org/10.6084/m9.figshare.21753551.v4[28].

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

## Acknowledgements

This research was supported by the National Science Foundation (NSF—1941344, S.R.) and Defense Advanced Research Projects Agency (DARPA—W911NF-18-2-0032, M.R.). The content is solely the responsibility of the authors and does not necessarily represent the official views of NSF or DARPA.

## Author contributions

A.A.A. and K.O. performed the experiments. A.A.A., J.K.R., B.M.R., E.K., M.R., and S.R. analyzed the results. J.K.R., B.M.R., E.K., M.R., and S.R. provided direction and oversight. A.A.A., B.M.R., J.K.R., and S.R. drafted the manuscript. All authors provided notes and edits to the manuscript.

## Competing interests

The authors declare no competing interests.
