## [Peer Review File · Nature Communications]

Digital Circuits and Neural Networks Based on Acid-Base Chemistry Orchestrated by Robotic Fluid HandlingREVIEWER COMMENTS

Reviewer #1 (Remarks to the Author):

The manuscript: "Computing with Acid-Base Reactions" presents the idea of computation with information coded in pH of the solution. Information coding is done with concentrations of selected acid and base. The information processing occurs with the acid-base neutralization reaction. The authors introduce a clever method for information coding with a pair of containers with solutions (dual-rail encoding). This is done to implement the negation operation, which can be performed by changing the order of containers. The idea is interesting, but I do not think the manuscript describes a significant step towards efficient chemical computation. Moreover, in my opinion, the section on digit recognition has not been clearly presented, and it is hard to understand. I do not recommend the publication of the manuscript in the present form.

Here are my specific comments:

- Abstract " encode binary information and perform majority and negation operations, entirely at the molecular level ", I disagree with this statement. Information is coded in the concentration of ions in the system of sufficient optical density to recognize the change in solution color.
- The system is not autonomous because it requires a mechanical manipulator to exchange the order of solutions in negation operation. If we allow for mechanical operations, then why not code binary operations with a single solution that will be replaced by the opposite one in the negation?
- I can see problems with information flow if gates are linked together. In most logic operations, the output solution is diluted if compared to input ones. Thus, in my opinion, it should be replaced to be used for the following operations.
- It is not directly said in the manuscript, but I presume the authors obtained a neural network for character recognition and implemented chemical operations to operate it. What was the network? How large was the training database for this network? How many elements were used for its validation? How many experiments were performed to obtain the network match?
- Figure 5 is unclear. As I understand, Fig 5a shows an example of digit 1 presented as 16x16 bit image. What is the relationship between 5a and 5b? Do pixels in 5a translate into color dots in 5b? How does 5c follow from 5b? How the final output is obtained from 5c, and what does it mean?
- p. 15 - Authors say that the time to write the encoded image was 40 minutes. But how long does it take to classify it? I would like to see such data for all networks mentioned in the tables.
- What is the meaning of the average accuracy given in tables? The compared networks seem to be completely different.
- Fig 6 - there are more colors used to mark dots, not only yellow and blue. What is their meaning?
- p.15 What was the error of pipetting 200 nL?

Reviewer #3 (Remarks to the Author):

In this work the authors explain how to use acid base reaction to perform digital logic operations such as inversion and majority functions, and also to perform pattern recognition against the NMIST dataset. Overall I like the simplicity of their approach. I Googled and searched on a few other databases to see if I could find something similar using acid and base reactions, but I couldn't beyond the paper I comment at the very end.

I think the paper is well written and it is clear to understand. Perhaps there should be a bit more of information in the figures. The experimental setup seems good, and their benchmark of their results seems also good and rigorous. The methods also contain enough information to reproduce the results.

I will first make some comments as I read the paper, and I will discuss and analyse the results at the end:

Abstract: It mentions “entirely at the molecular level”. Which I disagree. While it is true that acid base reactions work at the molecular level, that’s not how it has been used in this research. For example table 1 says that their lower volume is 50ml, while their biggest volume is 200 ml. So in order to flip from 0 to 1, or the other way around, they need an incredible big number of molecules. This is not a molecular computer, but more like a bulk or reservoir computer. They don’t have control over individual acid or base molecules, and operate individually. But they need a large number.

Intro: The criticism that silicon computing is almost at its max capacity and hence alternative computing architectures are needed, I disagree. This has been said for many years, and they are still the most powerful architecture, not only that they also get the biggest performance gains. I think overall this angle is wrong. I like more the last sentence :“Chemical computing approaches are moreover compatible with aqueous or extreme conditions, or more specialized applications such as soft robotics.” I think this angle is better and should be more explored in the intro, because I don’t think chem computer will ever reach the levels of current digital computers. This is a minor comment since a lot of unconventional computing papers use a similar angle. It is just my opinion.

In the intro: “Additionally, they can be easily realized without the need for expensive equipment or sophisticated procedures.” This is true, but working with very high or low ph is not safe. I don’t know if this is discussed later on, but it should perhaps. This is just a very minor comment. After reading the whole paper, to do the liquid handling, they use a robot “Echo 550, Labcyte” Google says that a used version of this robots costs around \$7000. This is not exactly cheap as the authors claim in the discussion. I understand that acid and bases are very cheap chemicals, but you also should consider all the infrastructure needed.

The paper seems to be based on “dual rail encoding”. But I’ve never heard about this before, and it is never introduced or explained. This should be developed. I had to google this myself. The negation operation in this paper is explained as: “the negation operation by exchanging the two solutions during the reaction to represent the inverted data”. This seems very far away from the molecular level. It seems that physically the solutions are swapped. I guess this is done mechanically or manually. If I am understanding this well, this has nothing to do with chemistry. I think the authors should clarify how this is done. Is it the robot handling the liquid?

Figure 1 should probably be joined with Figure 2. The abstract speaks about microfluidic devices, and I would have like to see this in Figure 1 (or perhaps figure 2). Figure 1 also appears before Figure 2, but in the text, Fig 2 is referenced first.

After reading the whole paper, probably the microfluidic sentence from the abstract should be removed.

Figure 2 is a bit puzzling. I don’t follow how the addition operation is performed, or how a NN works with their scheme. Is it just adding liquids together and checking the resulting ph? I don’t get how the NN works at all. “modelled as a sequence of acid-base reactions where a range of inputs is supported by diluting the input solution with a water ratio corresponding to the encoded input value.” No idea what this means, or why the second input has red arrows. Part C, the mapping, is easier to follow.

“We use a pH indicator to read out the final computation results based on the pH of each result.” I am not sure about what the authors mean here. So they have a well plate with a large number of wells. These wells are paired to follow their dual rail encoding. If it is positive they put one acid droplet in one, and an based droplet in the other. My question is, is the ph indicator used individually on each well? And how is this operation performed?

“After training weights, we model our system as shown in Figure 5.” How are the weights trained? In a digital computer? In the methods it seems it is all digitised.

I would also like to know some time bechmarking. From the moment you put the input image, to the moment it classifies it, how long it takes?

Overall thoughts about this research:

My main concern is how the inversion operation is performed. It seems like it has nothing to do with the chemistry. I also have overall concerns about how much the chemistry is doing, and how much the robot + digital computer are doing. I don't think the chemistry here is doing much.

After reading the paper, while I understand how the image classification is performed using the robot and chemistry, I don't understand if the digital circuit part was fully implemented in the lab, or if it was simulated. I guess they did in the lab, because there are sentences like "we manually replace the intermediate results with fresh acid or base solutions,..." but they don't know much data or results about this. Overall I would like to see more pictures of the actual robot, results and experiments.

This paper made me think a lot about this very recent paper:

<https://pubs.rsc.org/en/content/articlelanding/2021/sc/d0sc05860b>

Checking the author lists I see that 5 out of 8 of the authors are in both papers.

In this Chem Sci paper the authors also used a well plate, where they put some chemistry, and then they perform image processing. As the authors say in this paper, what the chemistry is actually doing is being some sort of activation function in the NN. And I think in this paper that we are reviewing, the chemistry is doing the same. The weights were calculated digitally. The robot places the chemistry in the right volumes, the robot then mixes them together. Really, the chemistry is just being an activation function.

Both this paper being reviewed and the one linked just now are quite similar in their overall idea. The authors should show how their approach is better so that it deserves publication in a better journal.

Dear Reviewers,

We appreciate your constructive comments and suggestions. Your reviews were thorough and the feedback insightful. We have strengthened our arguments and clarified many ambiguous points based on your comments. We feel that we have a much stronger paper as a result.

Several comments by the reviewers pertained to our encoding of information. Recall that our paper introduces a systematic method for performing digital computation with acid-base reactions. We introduce a so-called “dual-rail” encoding to perform a negation operation. It solves a significant challenge for molecular computing: how to represent negative values with concentrations. (Evidently, a concentration cannot become negative.) A significant conceptual observation in our paper is that the complementary nature of acid and bases forms a natural basis for a dual-rail encoding: it allows them to act as opposing values, with the excess or deficit on one side of the pair representing the *magnitude* of the value, and the direction – excess or deficit – the *sign* of the value. Using our dual-rail encoding, the appropriate side of the dual-rail pair can be used as an input for the following reaction allowing us to use value or its negation according to the targeted functionality. We show how our system can be used to design digital circuits. Additionally, we show the design and the experimental results for running a neural network digit classification using our system.

To address your comments, our main updates involve:

- 1) Explaining better how dual-rail encoding works and our rationale for using it.
- 2) Updating the figures to include more clarifying details.
- 3) Providing more details about the experiments.
- 4) Clarifying the neural network classifier’s objective, flow details, figures, and evaluation details.
- 5) Adding a section and experiments about microfluidic devices.

Please find our responses below; we have also tried to quote the updated text when feasible directly. Our updates have the following format:

Black: Reviewer’s comments.

Blue: Our response to the comment.

Italic: The updated text from the revised manuscript.

Bold: The updated/revised section in the manuscript.

Red: The modified text in the revised manuscript.

Reviewer #1

The manuscript: "Computing with Acid-Base Reactions" presents the idea of computation with information coded in pH of the solution. Information coding is done with concentrations of selected acid and base. The information processing occurs with the acid-base neutralization reaction. The authors introduce a clever method for information coding with a pair of containers

with solutions (dual-rail encoding). This is done to implement the negation operation, which can be performed by changing the order of containers. The idea is interesting, but I do not think the manuscript describes a significant step towards efficient chemical computation. Moreover, in my opinion, the section on digit recognition has not been clearly presented, and it is hard to understand.

We thank the reviewer for viewing our work as “clever” and “interesting.” In the following, we attempt to address your comments and clarify your concerns.

Here are my specific comments:

1. Abstract " encode binary information and perform majority and negation operations, entirely at the molecular level ", I disagree with this statement. Information is coded in the concentration of ions in the system of sufficient optical density to recognize the change in solution color.

We agree. We wanted to underscore that we rely on chemical reactions between acids and bases for our computations instead of just using ions for storage. We overstated the case. Our system is not *entirely* molecular, as you point out. We edited this statement in the abstract.

2. The system is not autonomous because it requires a mechanical manipulator to exchange the order of solutions in negation operation. If we allow for mechanical operations, then why not code binary operations with a single solution that will be replaced by the opposite one in the negation?

We appreciate this comment. First, we would like to explain the motivation for our dual-rail methodology.

Negating values in chemical computations has, in general, been a long-standing challenge for the molecular programming community. Unlike digital numbers, chemical concentrations are inherently positively-valued. This means that we cannot invert information stored in chemical concentrations without prior knowledge about those concentrations. For example, given a solution without prior knowledge of whether it is an acid or base, you cannot transform a base with a given pH into the acid with its opposite pH (a pH of 14 minus the initial pH) or an acid into a base with its opposite pH. Even so, the inversion operation is fundamental to computation and hence required for molecular programming. We thus devised a way to encode information in acids and bases and invert them via a “dual-rail” encoding. In dual-rail encoding, we overcome this challenge by storing and processing the data in its two forms, the original and inverted forms. Each point in the data comprises a pair of values that are complementary to each other where the operations are applied to the whole pair producing results also in dual-rail form. This allows us to represent positive and negative numbers by assigning one set of flasks to a positive rail and one to a negative rail, even though the concentrations in the flasks are

strictly positive. This is one way of resolving the seeming contradiction between positive concentrations and negative numbers in chemical computing.

Second, we should highlight that the reason we are able to use this system is due to the **chemical** nature of acid and bases, which allows them to behave in a complementary way within the context of our system. With this system, we can now model our computations, including the inverted inputs, without prior knowledge of whether they are positive or negative.

Thirdly, using a one rail solution would not be sufficient to model the negation operation. If we were to use a single solution, negation would require prior knowledge of the solution (whether it is an acid or base) in order to manually replace it with its negated value. In other words, whenever we have a value that we need to feed its negation to another input, we would need to **measure** its pH value, find or synthesize another solution with the opposite pH (14 minus the original value), then feed it to the next stage. However, in our approach, we don't need any prior knowledge of the value, and we don't need any measurement or manual intervention; instead, we just feed the other side of the dual-pair to the next stage to present the negated input.

Finally, addressing the point about our use of mechanical operations, the usage of the mechanical operations is only for conducting the chemical reactions, but the system itself is not dependent on the mechanical operation. Our computations are reliant on the acid-base reaction, their nature, and the presented dual-rail encoding. The behavior of acid-base reactions is analogous to the majority operation. The dual-rail encoding allows us to augment this behavior with a negation operation. The mechanical aspects are used to carry out these reactions according to the system we presented. We even added another section showing the same computation on another microfluidic device (**see Methods: Microfluidic Board section, Page 18**). If we take the neural network classifier for an example, it might be equivalent to saying, this sequence of **chemical** reactions will result in some pH (color) from which you can know if the digit is "0" or "1". More specifically, the negation operation is not a mechanical swap but rather a different connection terminal from the input. We might have misworded the concept by using mechanical terms such as exchanging or flipping, but in reality, mechanical manipulation is not really the core of the negation. These operations can all be pictured as just different connections: you choose which side of the input to connect to, the positive or the negative side (without manual flipping). Digital circuits use similar concepts where some circuits produce an output, and it is negation. For example, in the D-Flip Flop circuit shown below (a prevalent building block in most digital circuits), you can choose which version you want to feed to your next operation based on the function without any mechanical or manual exchange.

Accordingly, we have updated the revised manuscript (**Introduction, Page 3**) to provide more context about the dual-rail encoding and attempted to clarify the ambiguity regarding our use of mechanical operations.

Updated texts:

- *“In general, negation is a challenging operation to realize in chemical computation since reacting a solution to produce its complement is not feasible without prior knowledge of the solution.”*
 - *“For example, to our knowledge, there is no reaction that would both change an acid to base, and change a base to an acid.”*
 - *“We use the positive side when we need the positive logic, and the negative side when we need the inverted values. All outputs are automatically encoded in the dual-rail form which allows for cascading operations even when the function involves negating intermediate values. This concept is analogous to the logic used in some digital circuits that produce both the output and its complement, like the D Flip-Flop[.]”*
 - *“the presence of the complementary solution for the same value allows us to negate the encoded value by swapping it with the other side of the dual-rail encoding”*
3. I can see problems with information flow if gates are linked together. In most logic operations, the output solution is diluted if compared to input ones. Thus, in my opinion, it should be replaced to be used for the following operations.

Thank you for pointing this out. We agree. This is what we do, as stated at different points throughout the paper (e.g., **Introduction, Page 5; Discussion Section, Page 15**). We have updated the revised manuscript to emphasize this point (see **Results: Digital Circuits Section Page 8**), saying that *“As explained earlier, the outputs of intermediate stages may be diluted because of the water generated during neutralization; hence, we manually replace the intermediate results with fresh acid or base solutions after the first stage to avoid dilution issues. We discuss later a potential idea to resolve this issue.”* Additionally, we discuss potential solutions for this limitation such as using pH-sensitive hydrogel valves in the Discussion Section (see **Discussion Section, Page 15**).

4. It is not directly said in the manuscript, but I presume the authors obtained a neural network for character recognition and implemented chemical operations to operate it. What was the network? How large was the training database for this network? How many elements were used for its validation? How many experiments were performed to obtain the network match?

Thank you for your comment. The clarifications you ask for will help others reproduce our experiments. Our network consisted of two neurons for the 2-digit classification and three neurons for the 3-digit network. The training dataset had 5923 points for the first class, 6742 points for the second class, and 5958 points for the third class. The validation dataset had 980 points for the first class, 1135 points for the second class, and 1032 points for the third class. The simulation validation used the full validation dataset. We performed around 10 lab experiments for different samples from the validation set. We have updated the revised manuscript to include the requested details (**see Methods Section Pages 16 & 17**).

Updated texts:

- *“we trained a neural network model consisting of two fully-connected neurons on two digits (“0”, “1”)”*
 - *“For the final set of results, we extended our model by using three fully-connected neurons instead of two to classify binary images belonging to three different classes/digits (“0”, “1”, and “2”).”*
 - *“We ran our model using various parameters (image size, discretization levels, and the number of classes) across the full validation set”*
 - *“Data points for the digits “0”, “1”, and “2” were used for the network including 5923, 6742, and 5958 points for training, and 980, 1135, and 1032 points for validation, respectively.”*
 - *“10 Random samples from different validation images were selected and carried out physically by the liquid handler in the lab to verify the results.”*
5. Figure 5 is unclear. As I understand, Fig 5a shows an example of digit 1 presented as 16x16 bit image. What is the relationship between 5a and 5b? Do pixels in 5a translate into color dots in 5b? How does 5c follow from 5b? How the final output is obtained from 5c, and what does it mean?

Thanks for catching this. Fig 5a shows the digit being classified by the network. Fig 5b shows a stock of acids and bases that the robot used to encode the image into acids and bases in the next step. The acids and bases are distributed over multiple wells (instead of one big volume of acid and one big volume of base) to satisfy the robot's constraint on the maximum volume of fluid per well. The robot encodes the image in 5a into that in 5c (using the stocks of acids and bases in Fig 5b) based on the network

weights and the system we explained in the paper, where each pixel of value “1” would translate to an acid followed by a base, and a value “-1” would translate to a base followed by acid. If the pixel is associated with a negative weight, the order is reversed. Fig 5d is the final output of the network after pooling all of the dual-pairs of the encoded image, which corresponds to the output of each neuron in the network. If the neuron produces a(Base, Acid) output, this is equivalent to a zero (the neuron is off). If the neuron produces an(Acid, Base) output, this is equivalent to a one (the neuron is on). We have added more details to the revised manuscript and updated the figure to explain our workflow in more detail (see **Results: Digit Classifier Section, Page 12**).

Updated text:

- *“After training weights, we model our system as shown in Figure 5. Figure 5a) shows an example image from the validation dataset. Figure 5b) shows the initial (unencoded) stocks of acids and bases provided to the liquid handler to encode the input image. We start by encoding the image into its acid-base representation shown in Figure 5c) using the acid-base stock shown in Figure 5b). We encode the image as many times as the number of neurons (classes) in our model (we have two classes/neurons since we are classifying the digits “0” and “1”). When applying the weights, we either transfer the volumes in their original order for positive weights (multiply by 1) or the opposite order (multiply by -1) for negative weights. After encoding and applying the weights, we compute the output from each neuron to decide whether it is a “1” or “-1.” Finally, as shown in Figure 5d), for each neuron we pool (sum) all the positive (left) sides of the dual-rail to compute the positive side of the neuron's output, and all the negative (right) sides of the dual-rail to compute the negative side of the neuron's outputs. Finally, we add a pH indicator to the neuron outputs to determine whether the outputs are acidic or basic based on their color. The “1” neurons will have an output pair of (Acid, Base), while the “-1” neurons will have an output pair of (Base, Acid).”*

6. p. 15 - Authors say that the time to write the encoded image was 40 minutes. But how long does it take to classify it? I would like to see such data for all networks mentioned in the tables.

Thanks for your suggestion. We have added the table below of the experiments' run-times to the **Methods Section, Page 17**.

Image Size	2-Class Binary Image		2-Class 3-bit Image		3-Class Binary Image	
	Encoding	Pooling	Encoding	Pooling	Encoding	Pooling
8x8	13 min.	1 min.	13 min.	1 min.	19 min.	2 min.
12x12	29 min.	3 min.	29 min.	3 min.	44 min.	4 min.
16x16	52 min.	5 min.	52 min.	5 min.	78 min.	8 min.

28x28	158 min.	16 min.	158 min.	16 min.	237 min.	24 min.
-------	----------	---------	----------	---------	----------	---------

7. What is the meaning of the average accuracy given in tables? The compared networks seem to be completely different.

The given accuracy represents how well our chemical computation matches results of the digital network. The average accuracy represents how much our model matches the encoded neural networks over different input image sizes. We have added more clarification about the table to the revised manuscript (**see Results: Digit Classifier Section, Page 13**).

Updated text:

- "Our chemical-based classifier is able to match the theoretical results from the digital network with 99.61% similarity on average across different input sizes."
- "Our chemical-based classifier is able to match the theoretical results from the digital network with 98.94% similarity on average across different input sizes."
- "Our chemical-based classifier is able to match the theoretical results from the digital network with 98.35% similarity on average across different input sizes."

8. Fig 6 - there are more colors used to mark dots, not only yellow and blue. What is their meaning?

Thank you for your question. The different shades represent different concentrations/dilutions for the acids and bases based on the system we described. The base yellow and blue colors represent the initial acids and bases, respectively without dilution. A lighter yellow or a lighter blue would indicate a more diluted acid or base, respectively. We have added more clarifications to the revised manuscript (**see Methods section, Page 18**).

Updated text:

- *"A lighter yellow or blue represents a more diluted acid or base, respectively."*

9. p.15 What was the error of pipetting 200 nL?

The liquid handler has a maximum of 10% deviation from the target volume. We added this information to the revised manuscript (**see Methods section, Page 17**).

Updated text:

- *“The liquid handler has a maximum deviation of 10% from its target volume.”*

Reviewer #3

In this work the authors explain how to use acid base reaction to perform digital logic operations such as inversion and majority functions, and also to perform pattern recognition against the MNIST dataset. Overall I like the simplicity of their approach. I Googled and searched on a few other databases to see if I could find something similar using acid and base reactions, but I couldn't beyond the paper I comment at the very end.

I think the paper is well written and it is clear to understand. Perhaps there should be a bit more of information in the figures. The experimental setup seems good, and their benchmark of their results seems also good and rigorous. The methods also contain enough information to reproduce the results.

We thank the reviewer for viewing our work as “well written” and “clear to understand.” In the following, we attempt to address your comments and clarify any of your concerns.

I will first make some comments as I read the paper, and I will discuss and analyse the results at the end:

1. Abstract: It mentions “entirely at the molecular level”. Which I disagree. While it is true that acid base reactions work at the molecular level, that's not how it has been used in this research. For example table 1 says that their lower volume is 50ml, while their biggest volume is 200 ml. So in order to flip from 0 to 1, or the other way around, they need an incredible big number of molecules. This is not a molecular computer, but more like a bulk or reservoir computer. They don't have control over individual acid or base molecules, and operate individually. But they need a large number.

Thank you for noting this. As mentioned in this document, we wanted to highlight that we rely on chemical reactions between acids and bases to model our computations. We have updated the abstract accordingly and removed the words “molecular level.”

2. Intro: The criticism that silicon computing is almost at its max capacity and hence alternative computing architectures are needed, I disagree. This has been said for many years, and they are still the most powerful architecture, not only that they also get the biggest performance gains. I think overall this angle is wrong. I like more the last sentence :“Chemical computing approaches are moreover compatible with aqueous or extreme conditions, or more specialized applications such as soft robotics.” I think this angle is better and should be more explored in the intro, because I don't think chem

computer will ever reach the levels of current digital computers. This is a minor comment since a lot of unconventional computing papers use a similar angle. It is just my opinion.

Thanks for the remark, We agree with your perspective, and accordingly, we have updated the Introduction to eliminate its discussion of the limitations of semiconductor-based technologies and instead added more about the motivations for unconventional computing (**see Introduction, Page 1**).

Updated text:

- “Semiconductor devices have governed information processing over the past several decades. However, researchers have been exploring alternative methodologies for data processing, encouraged by their potential for greater power-efficiency, greater affordability, and ability to adapt to different environmental conditions that may be inhospitable or unnatural for conventional semiconductor technologies [].”
3. In the intro: “Additionally, they can be easily realized without the need for expensive equipment or sophisticated procedures.” This is true, but working with very high or low pH is not safe. I don’t know if this is discussed later on, but it should perhaps. This is just a very minor comment. After reading the whole paper, to do the liquid handling, they use a robot “Echo 550, Labcyte” Google says that a used version of this robots costs around \$7000. This is not exactly cheap as the authors claim in the discussion. I understand that acid and bases are very cheap chemicals, but you also should consider all the infrastructure needed.

Thanks for the feedback and the insight. As for the safety, there is no mandate to use any specific pH value; the pHs just need to be complementary (pH of acid = 14.0 - pH of the base). Theoretically, even acidic and basic solutions in regular house supplies would work if they have complementary pH values. Regarding the cost of the robot, first, we would like to highlight that the Echo 550 is a general-purpose liquid handler that is used by our team to carry out a multitude of different types of chemistry experiments and drug synthesis. Hence, the use of sophisticated equipment is by no means a prerequisite to running our experiments. The Echo 550 is just the platform we used to automate the process, but any other platform that does basic liquid transfers would still work well. In order to clarify this aspect, we have added a new experiment using a much cheaper microfluidic device (~€945) that performs similar operations, demonstrating that our chemistry is not specific to particular equipment (**see Methods: Microfluidic Board section, Page 18**).

4. The paper seems to be based on “dual rail encoding”. But I’ve never heard about this before, and it is never introduced or explained. This should be developed. I had to google this myself. The negation operation in this paper is explained as: “the negation

operation by exchanging the two solutions during the reaction to represent the inverted data". This seems very far away from the molecular level. It seems that physically the solutions are swapped. I guess this is done mechanically or manually. If I am understanding this well, this has nothing to do with chemistry. I think the authors should clarify how this is done. Is it the robot handling the liquid?

Thanks for the remark. There are some references in the paper to previous usages of similar concepts in other work, but we have added more explanation and context about the dual-rail encoding in the revised manuscript. Regarding the concern about the chemical aspect of the negation, as explained previously in this document: "Negating values in chemical computations has, in general, been a long-standing challenge for the molecular programming community. Unlike digital numbers, chemical concentrations are inherently positively-valued. This means that we cannot invert information stored in chemical concentrations without prior knowledge about those concentrations. For example, given a solution without prior knowledge of whether it is an acid or base, you cannot transform a base with a given pH into the acid with its opposite pH (a pH of 14 minus the initial pH) or an acid into a base with its opposite pH. Even so, the inversion operation is fundamental to computation and hence required for molecular programming. We thus devised a way to encode information in acids and bases and invert them via a "dual-rail" encoding. In dual-rail encoding, we overcome this challenge by storing and processing the data in its two forms, the original and inverted forms. Each point in the data comprises a pair of values that are complementary to each other where the operations are applied to the whole pair producing results also in dual-rail form. This allows us to represent positive and negative numbers by assigning one set of flasks to a positive rail and one to a negative rail, even though the concentrations in the flasks are strictly positive. This is one way of resolving the seeming contradiction between positive concentrations and negative numbers in chemical computing.

Second, we should highlight that the reason we are able to use this system is due to the **chemical** nature of acid and bases, which allows them to behave in a complementary way within the context of our system. With this system, we can now model our computations, including the inverted inputs, without prior knowledge of whether they are positive or negative.

Thirdly, using a one rail solution would not be sufficient to model the negation operation. If we were to use a single solution, negation would require prior knowledge of the solution (whether it is an acid or base) in order to manually replace it with its negated value. In other words, whenever we have a value that we need to feed its negation to another input, we would need to **measure** its pH value, find or synthesize another solution with the opposite pH (14 minus the original value), then feed it to the next stage. However, in our approach, we don't need any prior knowledge of the value, and we don't need any measurement or manual intervention; instead, you just feed the other side of the dual-pair to the next stage to present the negated input.

Finally, addressing the point about our use of mechanical operations, the usage of the mechanical operations is only for conducting the chemical reactions, but the system itself is not dependent on the mechanical operation. Our computations are reliant on the acid-base reaction, their nature, and the presented dual-rail encoding. The behavior of acid-base reactions is analogous to the majority operation. The dual-rail encoding allows us to augment this behavior with a negation operation. The mechanical aspects are used to carry out these reactions according to the system we presented. We even added another section showing the same computation on another microfluidic device (**see Methods: Microfluidic Board section, Page 18**). If we take the neural network classifier for an example, it might be equivalent to saying, this sequence of **chemical** reactions will result in some pH (color) from which you can know if the digit is “0” or “1”. More specifically, the negation operation is not a mechanical swap but rather a different connection terminal from the input. We might have misworded the concept by using mechanical terms such as exchanging or flipping, but in reality, mechanical manipulation is not really the core of the negation. These operations can all be pictured as just different connections: you choose which side of the input to connect to, the positive or the negative side (without manual flipping). Digital circuits use similar concepts where some circuits produce an output, and it is negation. For example, in the D-Flip Flop circuit shown below (a prevalent building block in most digital circuits), you can choose which version you want to feed to your next operation based on the function without any mechanical or manual exchange.

Accordingly, we have updated the revised manuscript (**Introduction, Page 3**) to provide more context about the dual-rail encoding and attempted to clarify the ambiguity regarding our use of mechanical operations.”

Updated texts:

- *“In general, negation is a challenging operation to realize in chemical computation since reacting a solution to produce its complement is not feasible without prior knowledge of the solution.”*

- *“For example, to our knowledge, there is no reaction that would both change an acid to base, and change a base to an acid.”*
 - *“We use the positive side when we need the positive logic, and the negative side when we need the inverted values. All outputs are automatically encoded in the dual-rail form which allows for cascading operations even when the function involves negating intermediate values. This concept is analogous to the logic used in some digital circuits that produce both the output and its complement, like the D Flip-Flop[.]”*
 - *“the presence of the complementary solution for the same value allows us to negate the encoded value by swapping it with the other side of the dual-rail encoding”*
5. Figure 1 should probably be joined with Figure 2. The abstract speaks about microfluidic devices, and I would have like to see this in Figure 1 (or perhaps figure 2). Figure 1 also appears before Figure 2, but in the text, Fig 2 is referenced first.

Thanks for catching this. We have merged the two figures and updated the captions and references accordingly (**see Introduction, Page 4**). For the microfluidic device, we are referring to the transfers of the liquids that perform the chemical reaction to model our system; we removed the word microfluidic to avoid confusion. We have also added an experiment using a microfluidic device to perform similar computations as an example (**see Methods: Microfluidic Board section, Page 18**).

6. After reading the whole paper, probably the microfluidic sentence from the abstract should be removed.

As explained before, the computations are agnostic to the platform used as long as it can perform the fluidic transfers. Our approach can be implemented on any device that can handle the transfer of fluids as explained. We have added an experiment using a microfluidic device to perform similar computations as an example (**see Methods: Microfluidic Board section, Page 18**).

7. Figure 2 is a bit puzzling. I don't follow how the addition operation is performed, or how a NN works with their scheme. Is it just adding liquids together and checking the resulting ph? I don't get how the NN works at all. “modelled as a sequence of acid-base reactions where a range of inputs is supported by diluting the input solution with a water ratio corresponding to the encoded input value.” No idea what this means, or why the second input has red arrows. Part C, the mapping, is easier to follow.

Thanks for pointing this out. The addition is mixing the liquids that are on the same side together (left or right) and the output is the resultant pH of the dual-rails (acid-base or base-acid). The NN is modeled by combining the encoded dual-rails in the original order

for positive weights, and in reverse order for negative weights. The dilution is only used if we want to encode more values than 1 and -1. The red arrow highlights the reversed order that corresponds to the negative weight. We have updated the revised manuscript to include all of these details in an attempt to explain our workflow more clearly (**see Introduction, Page 4 & 5**).

Updated text:

- *“All of the positive (left) sides of the inputs are mixed together to represent the positive side of the output. Similarly, all of the negative (right) sides of the inputs are mixed together to represent the negative side of the output resulting in an acidic solution ($pH < 7.0$) on the positive side and a basic solution ($pH > 7.0$) on the negative side, which correspond to an output of “1”. Additionally, for applications that can benefit from having a wider range of inputs other than “0” or “1”, we can map the input values after discretization into pH values as shown in Figure 1b. We can notice that the values are not evenly distributed due to the logarithmic nature of the pH scale. We explain in a later section the methodology we use for choosing the appropriate mapping.”*
 - *Figure 5c depicts a neuron (a building block for neural networks) modeled as a sequence of acid-base reactions where a range of inputs is supported by diluting the input solution with a water ratio corresponding to the encoded input value as explained in later sections. Similar to the addition/subtraction, all the positive sides of the inputs are mixed together to represent the positive side of the output, while all the negative sides of the inputs are mixed together to represent the negative side of the output. The red lines highlight using the reversed order of input to use its complemented value to apply the negative weight.} Finally, Figure 1d shows a full image of the digit “0” encoded as a sequence of dual-rail values to be processed through our acid-base computations.”*
8. “We use a pH indicator to read out the final computation results based on the pH of each result.” I am not sure about what the authors mean here. So they have a well plate with a large number of wells. These wells are paired to follow their dual rail encoding. If it is positive they put one acid droplet in one, and an based droplet in the other. My question is, is the ph indicator used individually on each well? And how is this operation performed?

Thanks for highlighting this. We have elaborated on our description of this experiment in the revised manuscript (**see Results: Data encoding Section, Page 7**). The pH indicator is only used on the final output which is what we are interested in reading (the final neuron output, like Figure 5d). The colors in the figures are just to demonstrate the acids and bases in the different wells. In the lab experiments, the solutions are usually transparent until the last step. After the final pooling step, the pH indicator is only added to the final output wells to know whether the solutions they contain are acids or bases. We can add the pH indicator to all the wells (including the intermediate stage) but

doesn't provide any useful information other than a visual representation of the encoded image. We add the indicator manually since the output usually involves only 4-6 wells, but it is very trivial to automate it with the liquid handler if needed.

Updated text:

- *“Finally, we add a pH indicator to the final outputs of the computation to read the results. If the indicator measures an (Acid, Base) pair, this signifies that the output is a “1”, while a (Base, Acid) pair would indicate “-1/0.””*

9. “After training weights, we model our system as shown in Figure 5.” How are the weights trained? In a digital computer? In the methods it seems it is all digitised.

Great observation. Neural networks have two phases: the training phase (which usually happens one time on a well-known dataset with the known result), and the inference phase happens after that by using the trained weights to infer or classify any new input. Training, which is usually a one-time process, is sensitive to floating-point precision and requires intensive computation resources and time. This type of computation is not feasible realistically within the chemical context we are providing. However, most applications or platforms (such as edge devices) delegate the training phase to other computation platforms with intensive resources and only use the trained model for inference. Similarly, our system is currently designed for inference (classification) and not training. Hence, while the weights are trained on a digital computer once, the classification/inference is done through the acid-base chemistry of our system, not through a digital platform. We explain these aspects and highlight that we are only performing inference in the revised manuscript (**see Results: Digit Classifier Section, Page 9**).

Updated text:

- *“Neural networks have two phases: a training phase and an inference phase. The training phase involves using a ready dataset to find optimal weights that generate the desired solution with high accuracy. The inference phase uses the pre-trained weights to find the solution for new inputs outside the dataset. Our work focuses on the inference phase based on pre-trained weights which is the main phase in practical applications.”*

10. I would also like to know some time benchmarking. From the moment you put the input image, to the moment it classifies it, how long it takes?

Thank you for the suggestion. We have added the table below of the experiments' run-times to the revised manuscript (**see Methods Section, Page 17**).

Image Size	2-Class Binary Image		2-Class 3-bit Image		3-Class Binary Image	
	Encoding	Pooling	Encoding	Pooling	Encoding	Pooling
8x8	13 min.	1 min.	13 min.	1 min.	19 min.	2 min.
12x12	29 min.	3 min.	29 min.	3 min.	44 min.	4 min.
16x16	52 min.	5 min.	52 min.	5 min.	78 min.	8 min.
28x28	158 min.	16 min.	158 min.	16 min.	237 min.	24 min.

Overall thoughts about this research:

11. My main concern is how the inversion operation is performed. It seems like it has nothing to do with the chemistry. I also have overall concerns about how much the chemistry is doing, and how much the robot + digital computer are doing. I don't think the chemistry here is doing much.

.
Thanks for the remark, re-iterating what we mentioned above, and adding more clarifications:

1. Negating values in chemical computations has, in general, been a long-standing challenge for the molecular programming community. Unlike digital numbers, chemical concentrations are inherently positively-valued. This means that we cannot invert information stored in chemical concentrations without prior knowledge about those concentrations. For example, given a solution without prior knowledge of whether it is an acid or base, you cannot transform a base with a given pH into the acid with its opposite pH (a pH of 14 minus the initial pH) or an acid into a base with its opposite pH. Even so, the inversion operation is fundamental to computation and hence required for molecular programming. We thus devised a way to encode information in acids and bases and invert them via a "dual-rail" encoding. In dual-rail encoding, we overcome this challenge by storing and processing the data in its two forms, the original and inverted forms. Each point in the data comprises a pair of values that are complementary to each other where the operations are applied to the whole pair producing results also in dual-rail form. This allows us to represent positive and negative numbers by assigning one set of flasks to a positive rail and one to a negative rail, even though the concentrations in the flasks are strictly positive. This is one way of resolving the seeming contradiction between positive concentrations and negative numbers in chemical computing.

2. We should highlight that the reason we are able to use this system is due to the **chemical** nature of acid and bases, which allows them to behave in a complementary way within the context of our system. With this system, we can now model our computations, including the inverted inputs, without prior knowledge of whether they are positive or negative.

3. Addressing the point about our use of mechanical operations, the usage of the mechanical operations is only for conducting the chemical reactions, but the system itself is not dependent on the mechanical operation. Our computations are reliant on the acid-base reaction, their nature, and the presented dual-rail encoding. The behavior of acid-base reactions is analogous to the majority operation. The dual-rail encoding allows us to augment this behavior with a negation operation. The mechanical aspects are used to carry out these reactions according to the system we presented. We even added another section showing the same computation on another microfluidic device (**see Methods: Microfluidic Board section, Page 18**). If we take the neural network classifier for an example, it might be equivalent to saying, this sequence of **chemical** reactions will result in some pH (color) from which you can know if the digit is “0” or “1”. More specifically, the negation operation is not a mechanical swap but rather a different connection terminal from the input. We might have misworded the concept by using mechanical terms such as exchanging or flipping, but in reality, mechanical manipulation is not really the core of the negation. These operations can all be pictured as just different connections: you choose which side of the input to connect to, the positive or the negative side (without manual flipping).
4. To summarize:
 - a. Our approach relies on the chemical properties of acid-base reactions as a method to model a digital computation.
 - b. The dual-rail encoding is not a mechanical operation but rather a format that captures the value and its negation at the same time, allowing us to model the negation operation without prior knowledge of the operation’s output. Dual-rail encoding is feasible because of the chemical nature of acids and bases within our context.
 - c. Negation is not mechanical exchange, it is a choice of which side to feed to your next input.
 - d. For the neural network classifier, as explained above, we only target the inference phase (the phase targeted by all practical applications), and the classification is done using chemical reactions.
 - e. The robotic agent is not a building block of any part of our system; instead, it's the platform that we chose to carry out the chemical reactions modeled by our system.

Updated texts:

- *“In general, negation is a challenging operation to realize in chemical computation since reacting a solution to produce its complement is not feasible without prior knowledge of the solution.”*
- *“For example, to our knowledge, there is no reaction that would both change an acid to base, and change a base to an acid.”*
- *“We use the positive side when we need the non-complement form, and the negative side otherwise. All presented operation outputs are also in the form of dual-rail encoding, which allows for cascading operations even when the function needs negated intermediate values. This concept is analogous to the logic used in some digital circuits like the D Flip-Flop [] that produce both the output and its complement.”*
- *“the presence of the complementary solution for the same value allows us to negate the encoded value by swapping it with the other side of the dual-rail encoding”*

12. After reading the paper, while I understand how the image classification is performed using the robot and chemistry, I don't understand if the digital circuit part was fully implemented in the lab, or if it was simulated. I guess they did in the lab, because there are sentences like “we manually replace the intermediate results with fresh acid or base solutions,...” but they don't know much data or results about this. Overall I would like to see more pictures of the actual robot, results and experiments.

We performed the digital circuit experiment through simulation, but it is very straightforward to perform in the lab since it involves very few transfers compared to the neural network classifier. We have added a figure and a paragraph showing and explaining the simulation for the 2-Bit Decoder (**see Results: Digital Circuits Section Page 10**).

Updated text:

- *“Finally, Figure 4 presents the simulation of the decoder circuit using the liquid handler. Figure 4a) shows the truth table for the 2-Bit decoder circuit, Figure 4b) shows the initial acid and base stocks used to encode the data, Figure 4c) shows the first level of the circuit that includes the original input and its complements in addition to acid and base stocks to be used as a bias for the AND gates, and Figure 4d) shows the final outputs of the circuits after applying the AND gate functionalities according to the circuit design.”*

13. This paper made me think a lot about this very recent paper:

<https://pubs.rsc.org/en/content/articlelanding/2021/sc/d0sc05860b>

Checking the author lists I see that 5 out of 8 of the authors are in both papers.

In this Chem Sci paper the authors also used a well plate, where they put some chemistry, and then they perform image processing. As the authors say in this paper,

what the chemistry is actually doing is being some sort of activation function in the NN. And I think in this paper that we are reviewing, the chemistry is doing the same. The weights were calculated digitally. The robot places the chemistry in the right volumes, the robot then mixes them together. Really, the chemistry is just being an activation function. Both this paper being reviewed and the one linked just now are quite similar in their overall idea. The authors should show how their approach is better so that it deserves publication in a better journal.

We appreciate your remark. Our current approach is significantly different from that described in the referenced paper. As the reviewer describes, the referenced paper focuses on using autocatalytic reactions for the chemical thresholding of chemical perceptrons to realize non-linear operations needed for image classification. While there is a minor similarity in terms of using the concentration for data encoding, our approach is radically different from the referenced paper. At a high level, the referenced paper cannot deal with positive and negative weights, which is a major drawback in any computation; our approach can handle negative values using dual-rail encoding. Additionally, autocatalysis is a very complicated reaction and requires time calibration (and the experiment for the referenced paper takes ~8 hours+), while our approach is much more straightforward to design and execute. We also provide methods to encode more values than "0" and "1" through different dilutions. Moreover, the readout of the referenced paper relies on UV-Vis Spectroscopy which is much more complicated than a simple color test with a pH indicator. Finally, our methodology is not an activation function. Our work is about using the acids, bases, and their reaction as a means to model numerical computations while using the dual-rail to model the negation operation. If we were to point out an analogy with digital computations, then what we mostly model in the neural networks section is matrix multiplication (the main component of neural networks) between the encoded input image (as dual-rail) and the trained weights (positive or negative). As explained above, training and inference are two separate parts of a neural network. It is also common when developing an application that uses a neural network to just download a model with already trained weights and jump to the inference process right away. Similarly, in our system, we are only interested in the inference part, so we use trained weights to model our system and classify the input digit images accordingly. We updated the revised manuscript to highlight that we are only tackling the inference phase and not the training in this work (**see Results: Digit Classifier Section, Page 9**).

REVIEWER COMMENTS

Reviewer #1 (Remarks to the Author):

I have read the second version of manuscript: "Computing with Acid-Base Reactions" submitted by the authors. I reads better than the original submission, but most of the remarks of my first report can be applied to the second version. The section on digit recognition has not been clearly presented, and it is still hard to understand.

I do not think the manuscript describes a significant step towards efficient chemical computation. There is little of autonomous chemical computation in the presented approach. The method requires mechanical operations (swapping) and external monitoring of solutions (diluting). The plate seems to act as a piece of paper where the stage of information processing is recorded, but the actual information processing can be done in a reactor outside it.

I agree that the idea of dual-rail coding and construction of logic gates is interesting for specialists in unconventional computation. But I do not think the manuscript in the present form can focus the attention of a wide audience.

And here are my specific comments:

- Page 6. I believe the idea of binary coding (-1,1) and three-state coding (-1,0,1) do work. However, due to dilution, the operations on elements of multiple state coding can be non-trivial, eg.: $2-1 = 1$
- Fig.2 The + sign has no sense in the definition of inversion.
- Fig 4. I presume the "computation" has been done outside the plate and Fig. 4d shows its final result.
- p. 11. The authors say "we trained a network model consisting of two neurons". Could the authors show this trained network? It is impossible to understand Fig. 5 without knowing what it is supposed to do.
- p. 17 what does it mean "2-class classification"?
- p.18 I do not understand the section on microfluidics. Is it simulation or experiment? What was the operation performed? Was it the average acidity of all 49 positive sites and the average acidity of all 49 negative sites? If so what kind of information on the image does it give?

Reviewer #3 (Remarks to the Author):

First of all I would like to say that the authors did an excellent job answering our comments and modifying the manuscript. The answers and changes were clear.

Checking the comments from the other referee, I see we both focused on the mechanical part of the "computation". The authors tried to address this point, but I still think that the robot is an indispensable part of the computation. The chemistry is only storing data, really. The logic is done in-silico (as the author say, the NN was pre-trained in a computer), and you still need pH probes, a robot to change the connections, and so. I don't think the chemistry is "computing" anything. The paper is more about how you can use acid and bases and dual-rail encoding to store data.

Something I liked about the other very similar paper the authors published (<https://pubs.rsc.org/en/content/articlelanding/2021/sc/d0sc05860b>) is that the authors focused very nicely the role of the chemistry = the task is image classification and the chemistry does a sort of activation function.

In the current manuscript, the title says "Computing with Acid-Base Reactions", and this is very far away from the reality. For example something they added in the abstract "orchestrated by a robotic fluid handling device".

There is nothing factually incorrect in the current version of the paper, but it is my opinion that the role of the chemistry should be better represented (starting by the title), and then the editor can decide if this new version interests to their journal.

Dear Reviewers,

We appreciate your constructive feedback and insights, and based on your notes, we have worked to clarify and elaborate on some aspects of the manuscript.

Several comments by the reviewers pertained to the role of chemistry in the computation. We would like to emphasize that these computations are only feasible because of the chemistry of the acid-base reactions and wouldn't be achievable otherwise. The reaction between acids and bases carries out the computation through neutralization which controls the concentration of the resultants which constitute our output values. We also introduce the novel approach of using dual-rail encoding to overcome the challenges of negation. The usage of our dual-rail encoding is only achievable because of the nature of acids and bases. If we were to use other chemicals, we would not be able to model the same computation or use the dual-rail representation since they won't have the same behavior or complementary nature.

The main updates to this version of the manuscript are:

- 1) We clarify that the reactions are chemical and not mechanical, while the role of the liquid handler is limited to transferring the inputs to their destinations.
- 2) Explaining the role of acid-base reaction and how our approach employs their chemistry to model computations.
- 3) Clarification of the role of the dual-rail logic and that it provides an approach to control the reaction through the order of the rails, not a mechanical operation.
- 4) Clarification of several ambiguous parts, such as the architecture of the neural network.

Please find our responses below; we have also tried to quote the updated text directly when feasible. Our updates have the following format:

Black: Reviewer's comments.

Blue: Our response to the comment.

Italic: The updated text from the revised manuscript.

Bold: The updated/revised section in the manuscript.

Red: The modified text in the first revision of the manuscript.

Magenta: The modified text in the second revision of the manuscript.

Reviewer #1

I have read the second version of the manuscript: "Computing with Acid-Base Reactions" submitted by the authors. It reads better than the original submission, but most of the remarks of my first report can be applied to the second version. The section on digit recognition has not been clearly presented, and it is still hard to understand.

Thank you; we have added a figure for the network and more explanation to the section to make it clearer:

- *For example, a digit classification network takes an input image and tries classifying it into one of ten classes (corresponding to each digit). The network generates one probability per class, indicating how likely the input image belongs to this digit class. For example, an output of (0.8, 0.1, 0.0125, 0.0125, 0.0125, 0.0125, 0.0125, 0.0125, 0.0125, 0.0125) indicates the input digit is classified as zero. This classification is done using a weighted sum of all input pixels of the images. The image can be presented in multiple formats: colored, grayscale, or black and white; we chose black and white. Additionally, the weight that is used for the summation can have different types; we chose binary weights, which are limited to "1" and "-1." [Page 9-10]*
- *A black pixel is represented by a base-acid pair, while a white pixel is represented by an acid-base pair. A positive weight indicates the pixel should be mixed in its original order, while a negative weight indicates the pixel should be mixed in the opposite order. [Page 11]*

I do not think the manuscript describes a significant step towards efficient chemical computation. There is little of autonomous chemical computation in the presented approach. The method requires mechanical operations (swapping) and external monitoring of solutions (diluting). The plate seems to act as a piece of paper where the stage of information processing is recorded, but the actual information processing can be done in a reactor outside it.

Thank you. While we do appreciate your concern, we do not see this the same way. The core contribution of this paper is to use the properties of the acid-base neutralization reaction to represent meaningful abstract computations. The role of the liquid handler is to transport input solutions to a destination on the plate outputs for the reaction to take place. The well plate simply holds the chemicals. The computation is done by the chemicals and how they react. If we were to replace the acids and bases with other chemicals, the computations would no longer work, because the complementary acid/base chemistry is central to the dual-rail concept and majority operations.

The dilution problem that would affect the cascading of multiple computations only shows up for certain examples, which require re-calibration at the intermediate stage to cascade; however, other examples are cascadable.

Our objective is to explore new chemical computation concepts and directions. The combination of complementary acid-base chemistry with storing complementary values via dual-rail logic enables us to perform a negation operation, which has been one of the major missing pieces in many prior examples of chemical computation. It is a bonus that the chemistry is universal, cheap, and simple to read out.

We elaborated on some relevant points in the manuscript, such as:

- *We explain how the nature of the acid-base reaction and the change in concentrations can model different computations. [Page 2]*

- *The reaction between acids and bases involves combining H_3O^+ cations from the acids with OH^- anions from the bases, resulting in an increase or decrease of the hydrogen ion H_3O^+ concentration based on the difference between the number of moles of H^+ and OH^- . When H_3O^+ and OH^- ions are pooled, they neutralize one another by forming water until only the more highly concentrated species remains. The resultant concentration indicates the output values of the computation. [Page 2]*
- *The role of the liquid handler is to transport input solutions to a destination on the plate outputs for the reaction to take place, analogous to how wires transport electricity from one place in an electronic circuit to another place. [Page 6]*
- *While some problems and input combinations are tolerant of these dilution constraints, other combinations may require re-calibration. [Page 5]*
- *Hence, we propose a dual-rail encoding for processed data that stores the data values as complementary pairs, e.g., (acid, base) or (base, acid), which enables the inversion operation by simply swapping the encoded solution during the reaction to represent the inverted data. [Page 3]*

And here are my specific comments:

- Page 6. I believe the idea of binary coding (-1,1) and three-state coding (-1,0,1) do work. However, due to dilution, the operations on elements of multiple state coding can be non-trivial, eg.: $2-1 = 1$

Thank you for your remark; our system only targets problems with an output of two classes (-1, +1) or three classes (-1,0,+1). Hence, the encoding shown works properly to get the expected output **sign** from the computation. For example, when $2-1$ is represented by our system, it will result in an output of the form (Acid, Base), which represents +1. So the presented encoding should always result in the correct output sign for any input combination at the first level of computation. The dilution problem that limits the cascading only shows up for specific input combinations that result in unbalanced outputs, which require re-calibration at the intermediate stage to cascade; some other input combinations are cascadable for more multi-level reactions.

We also added the following to the manuscript:

- *While some problems and input combinations are tolerant of these dilution constraints, other combinations may require re-calibration. [Page 5]*
- Fig.2 The + sign has no sense in the definition of inversion.

Thank you for pointing this out. We wanted to make all the representations have a uniform structure, but we removed the + sign accordingly to avoid confusion.

- Fig 4. I presume the "computation" has been done outside the plate and Fig. 4d shows its final result.

No, the computation (inference) was performed on the plate. This computation represents two fully-connected neurons with binary outputs. The neural network was trained off-line, and all inference computations were done on the plate. The dual-rail encoded chemical samples representing the image were transferred to the output wells, and the reactions in the output wells determined the outputs of the two neurons.

Neural network training and inference are two completely different steps in any model, whether in silicon or chemical computations. Training a neural network is a much more demanding task that is usually not feasible even on normal machines and requires training on processing farms with hundreds of powerful GPUs. Accordingly, most applications use a pre-trained model; we have added references for other computation paper that handles neural networks the same way, such as:

<https://www.nature.com/articles/nnano.2017.127>

<https://www.nature.com/articles/s41586-018-0289-6>

- p. 11. The authors say "we trained a network model consisting of two neurons". Could the authors show this trained network? It is impossible to understand Fig. 5 without knowing what it is supposed to do.

Thanks for your remark; the network is only two **fully connected neurons**. We added the following diagram for the network to make it clearer.

- p. 17 what does it mean "2-class classification"?

Thank you for your question. By 2-class classification, we mean a classification problem that has two possible outputs (e.g., classifying the image as 0 or 1). We added an explanation in the manuscript saying:

- *For example, a digit classification network takes an input image and tries classifying it into one of ten classes (corresponding to each digit). The network generates one probability per class, indicating how likely the input image belongs to this digit class. For example, an output of (0.8, 0.1, 0.0125, 0.0125, 0.0125, 0.0125, 0.0125, 0.0125, 0.0125, 0.0125) indicates the input digit is classified as zero. This classification is done using a weighted sum of all input pixels of the images. The image can be presented in multiple formats: colored, grayscale, or black and white; we chose black and white. Additionally, the weight that is used for the summation can have different types; we chose binary weights, which are limited to "1" and "-1." [Page 9-10]*
 - *Matching accuracy for 2-class, binary image network (classifying input image to digit zero, or digit one) [Page 14].*
 - *Matching accuracy for 3-class, binary image network (classifying input image to digit zero, digit one, or digit two) [Page 14].*
- p.18 I do not understand the section on microfluidics. Is it simulation or experiment? What was the operation performed? Was it the average acidity of all 49 positive sites and the average acidity of all 49 negative sites? If so what kind of information on the image does it give?

Thank you for your comment. We have decided to move this section to supplementary material to limit confusion. Based on earlier reviewer feedback, we added the microfluidic system simulation to illustrate that the approach can be performed on any liquid handling platform. The 7x7 image classification was a simulation because the board has volume limitations per well that hinders the execution of the reaction. However, we ran an actual physical experiment to model the operation of an AND logic gate to validate the system's feasibility. Based on our encoding, the outputs are the final pH values of all of the mixed wells. The objective of the experiment is to show that the system is not directly reliant on a specific fluidic handler but rather on any platform that can handle fluid transfers. We added more details to the supplementary material:

- *The final output is shown as the pH/color of the dual-pair of the droplet in the lower-left corner of the grid and the droplet in the lower-right corner, where acid-base would indicate +1, and base-acid would indicate -1/0.*

Reviewer #3

First of all I would like to say that the authors did an excellent job answering our comments and modifying the manuscript. The answers and changes were clear.

Checking the comments from the other referee, I see we both focused on the mechanical part of the "computation". The authors tried to address this point, but I still think that the robot is an indispensable part of the computation. The chemistry is only storing data, really. The logic is done in-silico (as the author say, the NN was pre-trained in a computer), and you still need pH probes, a robot to change the connections, and so. I don't think the chemistry is "computing"

anything. The paper is more about how you can use acid and bases and dual-rail encoding to store data.

We appreciate your feedback; the robot's sole objective is to transport input solutions to a destination on the plate outputs for the reaction to take place or to move the output of the reaction to the input of another, analogous to how wires transport electricity from one place in an electronic circuit to another place. The robot doesn't understand any concept of "swapping" but rather what is each source and destination. In other words, the computation is done purely through the acids and bases and how they react. The whole approach is based solely on acid-base chemistry, the dual-rail concept, how mixing acids and bases can act as a majority operation or weighted sum, and how different orders of the mixtures can represent negation. **For example, if we were to replace acid and bases with other chemicals (e.g., phenols, autocatalytic reactions..etc.), the computations won't be feasible, even in the presence of any robot/liquid handler, since we won't have the complementary nature for the dual-rail or the majority operation modeled throughout the reaction and the concentrations.**

Additionally, For the measurement, we only relied on the pH indicator, which shows colors easily identifiable by the naked eye. Hence, the measurement of any output doesn't require any sophisticated equipment other than noticing the color change after adding the indicator, which is a very cheap and fast measurement. Hence, the approach introduces unique techniques for conducting meaningful computations in chemistry and performing different operations and measurements.

For the NN part, as explained above, the computation (inference) was performed on the plate. Training and inference are two completely different steps in any model, whether in silicon or chemical computations. Training a neural network is a much more demanding task that is usually not feasible even on normal machines and requires training on processing farms with hundreds of powerful GPUs. Accordingly, most applications use a pre-trained model; we have added references for other computation paper that handles neural networks the same way, such as:

<https://www.nature.com/articles/nnano.2017.127>

<https://www.nature.com/articles/s41586-018-0289-6>

We elaborated on some of the mentioned points in the manuscripts, such as:

- *Neural networks have two phases: a training phase and an inference phase. The training phase is a computationally-intensive phase that involves using a ready dataset to find optimal weights that generate the desired solution with high accuracy. The inference phase uses the pre-trained weights to find the solution for new inputs outside the dataset. Hence, existing DNA-based work trains the system in a conventional computer and uses the DNA system for inference, and we follow the same protocol in our approach. Our work focuses on the inference phase based on pre-trained weights, which is the main phase in practical applications. [Page 10]*
- *We explain how the nature of the acid-base reaction and the change in concentrations can model different computations. [Page 2]*

- *The reaction between acids and bases involves combining H_3O^+ cations from the acids with OH^- anions from the bases, resulting in an increase or decrease of the hydrogen ion H_3O^+ concentration based on the difference between the number of moles of H^+ and OH^- . When H_3O^+ and OH^- ions are pooled, they neutralize one another by forming water until only the more highly concentrated species remains. The resultant concentration indicates the output values of the computation. [Page 2]*
- *The role of the liquid handler is to transport input solutions to a destination on the plate outputs for the reaction to take place, analogous to how wires transport electricity from one place in an electronic circuit to another place. [Page 6]*
- *While some problems and input combinations are tolerant of these dilution constraints, other combinations may require re-calibration. [Page 5]*
- *Hence, we propose a dual-rail encoding for processed data that stores the data values as complementary pairs, e.g., (acid, base) or (base, acid), which enables the inversion operation by simply swapping the encoded solution during the reaction to represent the inverted data. [Page 3]*

Something I liked about the other very similar paper the authors published (<https://pubs.rsc.org/en/content/articlelanding/2021/sc/d0sc05860b>) is that the authors focused very nicely the role of the chemistry = the task is image classification and the chemistry does a sort of activation function.

Thank you, we appreciate this. In the other paper, the valuable feature of the chemistry was its nonlinear transfer function; however, in that paper, all values and weights were positive numbers. In this paper, the important feature of the chemistry is the complementary nature of acid/base neutralization that map onto computations, including both positive and negative values. As we mentioned in an earlier response, the acid/base chemistry is integral to the system, and if we were to substitute different chemicals, the approach would no longer work.

In the current manuscript, the title says "Computing with Acid-Base Reactions", and this is very far away from the reality. For example something they added in the abstract "orchestrated by a robotic fluid handling device".

Thank you. As described earlier, the reaction between acids and bases is the basis of all the presented computations. The role of the liquid handler is to transport input solutions to a destination on the plate outputs for the reaction to take place, analogous to how wires transport electricity from one place in an electronic circuit to another place. Other fluid handlers (or even manual transfers) could have been used to implement the demonstration, and it would not change the core scientific contribution.

REVIEWER COMMENTS

Reviewer #1 (Remarks to the Author):

I have read the second version of the corrected manuscript.

First, I agree with the other that the role of chemistry in the presented "computation" is marginal. It is reduced to majority-type operations and to encoding the system state in mechanically controlled, chemistry-based RAM memory. Some classical chemical algorithms (Adleman DNA, prairie fire) are parallel. The one described in the manuscript is not and can be done sequentially using a few test tubes without a liquid handler. Here possible parallelism depends only on the mechanical construction of the pipetting device.

Second, in my opinion, the manuscript reads well at the beginning. However, the section "MNIST digit classifier using acid-base reactions" is chaotic and does not include information needed to follow the presented results. Let me be more precise:

- At the beginning of this Section, p.9 the authors refer to classification of all digits of decimal system based on a network with the output layer of 10 elements. Such a network is not considered in their research, so the information can confuse the reader because the manuscript is concerned with the classification of two {0,1} or three {0,1,2} symbols only.

- Figure 5 shows the 2-neuron network for the classification of 8x8 images into zero or one. The example does not give complete information about the network. How did the pixel position translate into the index of the input neuron? What were the weights of connections? What were the activation functions of the output neurons?

The authors should give this information at least for 8x8 image classification.

Moreover, it would help the reader if the authors define networks for 8x8 image classification of 3 - digits and for 3-bit grayscale coding.

- Figure 6b. In my opinion, the information on the location of stock solutions in a handler is not important for image processing.

- How does pH on the output neuron translate into the classification probability?

- The language of the Section on networks should be rechecked. There is a lot of jargon there; for example:

"The "1" neurons will have an output pair of (Acid, Base), while the "-1" neurons will have an output pair of (Base, Acid). "

or, more hard to understand:

"The mismatch between the simulated and experimental pHs arises from the computations whose outputs contain multiple "1" s instead of "1" for the classification and "-1" for the remaining neurons, which leads the output colors to be indistinguishable."

p. 13 Table 4 appears like a rabbit out of a hat. No information on 3-bit grayscale image coding and its classification are given in the Results section. Some information helping to understand what it means is long after Discussion, in the Methods section, at the end of the manuscript. Is the network for such classification the binary one?

p.18 I doubt if the data in Table 6 are correct.

In my opinion, for binary image encoding, it does not matter if the image represents a digit from {0,1} or from {0,1,2}; thus, the encoding times should be similar. On the other hand, 3-bit image coding should take more time because more different initial concentrations should be selected. Therefore, I think the authors should swap two rightmost sets of results

p.18. The authors should specify how many experimental tests of networks were done for each image size. As I read (p.16) "980, 1135, and 1032 points for validation" were used. The authors do not mention if the validation was in-silico only or both in-silico and in real experiments. If these input data were used in experiments, then they should perform over 3000 experiments for each image size. Keeping in mind that up to 4 hours are needed to encode a single digit (Table 6) it would take many years to obtain presented in Tables 3-5.

In conclusion: I do not recommend the manuscript in the present form for publication in Nature Communications.

Reviewer #3 (Remarks to the Author):

I do agree with a couple of things the other referee said:

"The plate seems to act as a piece of paper where the stage of information processing is recorded, but the actual information processing can be done in a reactor outside it."

"I agree that the idea of dual-rail coding and construction of logic gates is interesting for specialists in unconventional computation. But I do not think the manuscript in the present form can focus the attention of a wide audience."

I have spent too much reading the previous versions of the manuscript and the very long and verbose answers from the authors. I am afraid I don't have time for another lengthy round because I feel we are going around in circles.

I will only focus on one point of their answer:

"The robot doesn't understand any concept of "swapping" but rather what is each source and destination"

So, how is the swapping decided and implemented? Does the chemistry initiate some sort of reaction cascade which ultimate swaps the rails without any kind of interference from the robot, computer or electronics?

Dear Reviewers,

We greatly appreciate your constructive feedback and insights. Based on your notes, we have rewritten the manuscript to provide a clearer picture of our key contributions to the field and understanding of our methodology.

Several reviewer comments question how truly chemical our computations are. As highlighted in the updated manuscript, the novel dual-rail logic and pooling operations we describe are entirely predicated on the chemistry of acid-base reactions. Indeed, dual-rail logic relies upon being able to perform complementary reactions on the opposite rails, which can only be accomplished using complementary chemistries like the acid-base chemistry. As far as we know, no team has ever previously shown how dual-rail logic can be used to negate the generally positive chemical concentrations used to store information in small molecules. As a result, there have been remarkably few previous demonstrations of small molecule digital circuits and neural networks because of the essential role that negation plays in computation. While some aspects of our system require mechanical transfers, our core methodology - the storage of information in chemical concentrations, the pooling to perform summations, the complementarity to perform the dual-rail logic, and our overarching logic gate construction - is thus founded upon chemical reactions that are executed throughout the computation. We hope that we have more clearly articulated these points in the current manuscript.

The main updates to this version of the manuscript are:

- 1) Updated the introduction to highlight the critical role that chemistry assumes in our computations, and the main reactions driving our work.
- 2) Edited several sections to be more concise and provide a clearer understanding of our approach.
- 3) Provided more details and explanations about the neural network implemented in our system.
- 4) Updated the figures based on the feedback to add/remove the components highlighted by the reviewer.

Please find our responses below; we have also tried to quote the updated text directly when feasible. Our updates have the following format:

Black: Reviewer's comments.

Blue: Our response to the comment.

Italic: The updated text from the revised manuscript.

Bold: The updated/revised section in the manuscript.

Red: The modified text in the first and second revisions of the manuscript.

Magenta: The modified text in the third revision of the manuscript.

Reviewer #1

I have read the second version of the corrected manuscript.

First, I agree with the other that the role of chemistry in the presented "computation" is marginal. It is reduced to majority-type operations and to encoding the system state in mechanically controlled, chemistry-based RAM memory. Some classical chemical algorithms (Adleman DNA, prairie fire) are parallel. The one described in the manuscript is not and can be done sequentially using a few test tubes without a liquid handler. Here possible parallelism depends only on the mechanical construction of the pipetting device.

We appreciate your feedback. As we explained in the revised manuscript, our methodology is based upon the hybridization reaction now provided in the updated Introduction. We experimentally realized the majority function using chemical reactions since it can provide a universal representation for logic functions when paired with the negation operation (modeled in our dual-rail encoding). Indeed, chemistry performs the majority operation in just one mixing step, as compared with the many operations that need to be performed on a digital computer. In this sense, the chemical majority operation is performed in parallel. While our work does not highlight other forms of parallel computation because it is based on gates and neural networks that inherently require some level of sequential operations (from layer to layer, etc.), our focus was to demonstrate the modeling of universal computations and neural networks via small molecule chemical reactions rather than targeting specific problem sets like the ones presented in the referenced work. Universal computation as described here can ultimately be brought to bear on a wider class of problems than problem-specific methods.

Second, in my opinion, the manuscript reads well at the beginning. However, the section "MNIST digit classifier using acid-base reactions" is chaotic and does not include information needed to follow the presented results. Let me be more precise:

- At the beginning of this Section, p.9 the authors refer to classification of all digits of decimal system based on a network with the output layer of 10 elements. Such a network is not considered in their research, so the information can confuse the reader because the manuscript is concerned with the classification of two {0,1} or three {0,1,2} symbols only.

Thank you for the feedback. We provided the 10-neuron network as an example. We only used 2-3 neurons in our actual experiments. To avoid confusion, we removed the specification of the number of classes in the first paragraph:

- *Classification is a common task for neural networks to classify inputs (e.g., handwritten digit image) into a set of classes (e.g., the corresponding digit number). [Page 7]*
- Figure 5 shows the 2-neuron network for the classification of 8x8 images into zero or one. The example does not give complete information about the network. How did the

pixel position translate into the index of the input neuron? What were the weights of connections? What were the activation functions of the output neurons?

The authors should give this information at least for 8x8 image classification. Moreover, it would help the reader if the authors define networks for 8x8 image classification of 3 -digits and for 3-bit grayscale coding.

Thanks for pointing these ambiguities out. While we previously provided the source code and weights for all of our experiments with supplementary material of the submission, we have since updated the manuscript to much more explicitly include this information. In particular, our networks employed a Softmax activation function. We also updated the manuscript to indicate how the pixel values were translated into neuron inputs:

- *The pixels of the input image are flattened sequentially (left to right, and top to bottom) into a 1D vector. A copy of the 1D vector representation of the image is fed into each neuron. [Page 8]*

Additionally, we updated the supplementary material to include an image of the weights used for the 8x8 classification and the architecture of the 3-digit 3-bit grayscale images.

- Figure 6b. In my opinion, the information on the location of stock solutions in a handler is not important for image processing.

Thank you for your remark. Yes, the locations of the stock solution don't affect the state of the experiment. However, we thought it would enhance readers' understanding of our experiments to show the different processes that occur on the plates during the course of the experiments. We updated the figure and removed the stock solutions image to avoid confusion.

- How does pH on the output neuron translate into the classification probability?

Thank you again for your question. One of our main objectives was to simplify our measurements whenever feasible. Hence, our model relies on using the color of the indicator to determine the output of the neurons: (Base/Blue, Acid/Yellow) = Off, (Acid/Yellow, Base/Blue) = On, where the first chemical/color in the pair denotes what is present on the first rail and the second denotes what is present on the second rail. However, if we were to measure the exact pH value, how acidic or basic would correspond to how high the classification probability is. For instance, if the output of one neuron has the pH (0.0, 14.0), this would indicate almost 100% classification probability.

- The language of the Section on networks should be rechecked. There is a lot of jargon there; for example:

"The "1" neurons will have an output pair of (Acid, Base), while the "-1" neurons will have an output pair of (Base, Acid). "

or, more hard to understand:

"The mismatch between the simulated and experimental pHs arises

from the computations whose outputs contain multiple "1" s instead of "1" for the classification and "-1" for the remaining neurons, which leads the output colors to be indistinguishable."

Thank you for highlighting this unclear language. We rewrote the neural network section to be clearer and more concise, such as:

- *A neuron with an output pair (Acid, Base) indicates that the input image matches the digit this neuron is supposed to recognize. For instance, the first neuron is supposed to recognize the digit 0, while the second neuron is supposed to recognize the digit 1. In contrast, a neuron with an output pair of (Base, Acid) indicates that the input image does not match the digit this neuron is supposed to recognize. [Page 9]*
 - *The reason that there are minor differences between outputs from the simulation of the chemical experiment and the digital neural network, is that our methodology uses the output sign of the neuron to determine if it is on or off (a positive neuron output is on, while a negative neuron is off) while the digital circuit uses the maximum value instead. Hence, if the digital network has multiple positive outputs, they become indistinguishable in the chemical experiment since they produce the same color with the pH indicator.[Page 10]*
- p. 13 Table 4 appears like a rabbit out of a hat. No information on 3-bit grayscale image coding and its classification are given in the Results section. Some information helping to understand what it means is long after Discussion, in the Methods section, at the end of the manuscript. Is the network for such classification the binary one?

Thank you for your input. We have now added the relevant information to the manuscript:

- *For the second evaluation, we extended our experiment to classify 3-bit images (8 grayscale levels) instead of binary values. We use a network that is similar to the previous experiment but uses 3-bit values instead of binary ones. We discretized the input pixel values into eight different integers where each pixel value x is mapped to the integer: $\text{round}(8 \times x/255.0) - 4$. We then utilized the dilution equation explained before to represent the given integer. We re-trained the network with the updated representation. [Page 10]*
- p.18 I doubt if the data in Table 6 are correct. In my opinion, for binary image encoding, it does not matter if the image represents a digit from $\{0,1\}$ or from $\{0,1,2\}$; thus, the encoding times should be similar. On the other hand, 3-bit image coding should take more time because more different initial concentrations should be selected. Therefore, I think the authors should swap two rightmost sets of results

Good observation. The time to complete the experiment depends on the number/volumes of liquid transfers. As you can see from Figure 6b, we create a copy of the image for each neuron after applying its weights. For the 3-bit grayscale images, we require the same number of transfers, except that some of the acid/base transfers are substituted with water transfers for dilution to represent the 8 possible grayscale values

in 3-bit grayscale images. In contrast, the 3-class image adds overhead because transfers from applying the weights have to be performed three times because of the 3 neurons used in the classification. We included the reasoning in the updated manuscript:

- *The run-time of the experiment depends on the number/volume of liquid transfers needed to execute the experiment. The 2-class binary Image takes the same amount of time as the 3-bit grayscale variant since the transferred volumes are equivalent (the 3-bit grayscale variant only substitutes some of the transferred acid/base with water for dilution). On the other hand, the 3-class variant has an extra run-time overhead since the need to employ three neurons to classify three digits requires three sets of liquid transfers instead of two (the 3-class network employs three neurons, compared to the two neurons for the 2-class network). [Page 14]*

- p.18. The authors should specify how many experimental tests of networks were done for each image size. As I read (p.16) "980, 1135, and 1032 points for validation" were used. The authors do not mention if the validation was in-silico only or both in-silico and in real experiments. If these input data were used in experiments, then they should perform over 3000 experiments for each image size. Keeping in mind that up to 4 hours are needed to encode a single digit (Table 6) it would take many years to obtain presented in Tables 3-5.

Thank you for your remarks. The validation was done *in silico using a pH simulator, following typical neural network validation procedures*. To validate our chemistry, we classified a random sample from our validation dataset in our real experiments because, as you correctly note, running 3000 experiments was neither feasible nor would be needed to validate our weights. We also created a pH simulator that we used to computationally verify our reaction outcomes. We clarified these aspects in the manuscript:

- *The whole dataset was used for training and simulation in silico, while a random sample from the validation dataset was used for the experimental verification as described in the next section. [Page 13]*
- *In order to verify our results on the full dataset, an automated pH simulator was designed to simulate the reaction outcomes for the full dataset. [Page 13]*

Reviewer #3

I do agree with a couple of things the other referee said:

- "The plate seems to act as a piece of paper where the stage of information processing is recorded, but the actual information processing can be done in a reactor outside it."

"I agree that the idea of dual-rail coding and construction of logic gates is interesting for specialists in unconventional computation. But I do not think the manuscript in the present form can focus the attention of a wide audience."

Thank you for your feedback. We have updated our introduction to provide a more in-depth analysis of our methodology and the advantages it provides compared to other molecular computing approaches. We have also more clearly articulated the key role that chemistry assumes in our technique and the equations that underlie this chemistry. As highlighted in the updated manuscript, the novel dual-rail logic and pooling operations we describe are entirely predicated on the chemistry of acid-base reactions. Indeed, dual-rail logic relies upon being able to perform complementary reactions on the opposite rails, which can only be accomplished using complementary chemistries like the acid-base chemistry. As far as we know, no team has ever previously shown how dual-rail logic can be used to negate the generally positive chemical concentrations used to store information in small molecules. As a result, there have been remarkably few previous demonstrations of small molecule digital circuits and neural networks because of the essential role that negation plays in computation. While some aspects of our system require mechanical transfers, our core methodology - the storage of information in chemical concentrations, the pooling to perform summations, the complementarity to perform the dual-rail logic, and our overarching logic gate construction - is thus founded upon chemical reactions that are executed throughout the computation.

This work holds value for the wider molecular programming community because it opens the door to truly programming small molecules (many assume that only DNA/RNA can be programmed) by showing how multilayer circuits and neural networks that do useful computations can be realized using small molecule reactions. The inability to realize a chemical negation operation combined with limited means of cascading reactions has hampered progress in small molecule programming for decades. We believe that our approaches and logic will provide key ways of surmounting these challenges. To more directly address your suggestion that our reactions are not doing information processing and we are using our plates as pieces of paper that store information, consider what would happen if we removed: a) the majority operation performed by the acid-base mixtures or b) the complementary acid-base chemistry needed to perform the dual-rail gates we describe.

As described in the manuscript, the neutralization reaction performs a majority operation, which can be described as follows in digital logic:

$$Maj(x_{1\dots n}, y_{1\dots m}) = \begin{cases} 1 & \text{if } \sum_{i=1}^n x_i > \frac{n+m}{2} \\ 0 & \text{if } \sum_{i=1}^m y_i > \frac{n+m}{2} \end{cases}$$

The neutralization reaction can similarly be written as:

$$f(Acid_{1\dots n}, Base_{1\dots m}) = \begin{cases} Acid & \text{if } \sum_{i=1}^n Acid_i > \frac{n+m}{2} \\ Base & \text{if } \sum_{i=1}^m Base_i > \frac{n+m}{2} \end{cases}$$

If we removed the neutralization reaction, then we would have to compute all of the operations involved in the typical digital majority function. This would not be possible to do using a liquid handling system, and would involve numerous operations on a classical computer; the neutralization reaction only necessitates one mixing as opposed to many digital operations. Furthermore, if we were to remove the complementarity of the acid-base chemistry, we would not be able to perform opposite reactions on our dual-rails, which would eliminate these rails' dual nature and foil our logic. Thus, our reactions are critical to our information processing capabilities and our wells serve as more than just a piece of paper.

I will only focus on one point of their answer:

- "The robot doesn't understand any concept of "swapping" but rather what is each source and destination"

So, how is the swapping decided and implemented? Does the chemistry initiate some sort of reaction cascade which ultimately swaps the rails without any kind of interference from the robot, computer or electronics?

Thanks for pointing this out; we acknowledge that the wording might not have been accurate. What we were referring to here is that the robot's job is purely to act like the wiring in digital circuits. Each source well has a corresponding destination, just like an electronic wire has a source and destination. The robot transfers each source well to its set destination where the reactions take place. Because chemical reactions that are needed to realize our operations can't yet be performed in the same space at the same time, we do need robotics to segregate the reactions in space or microfluidics to potentially separate them in space and time, unless more challenging orthogonal chemistries are employed.

REVIEWERS' COMMENTS

Reviewer #1 (Remarks to the Author):

The present (4th?) version of the manuscript is more informative than the previous one; still, many parts should be improved before I can accept the manuscript.

In my opinion, the described chemical classifier can be interesting to a wide audience if it is based on a real working setup. I presume most of the results included in the present version of the manuscript were obtained using numerical simulations of the dual rail processes. My request for precise information on how many experimental tests were made:

"The authors should specify how many experimental tests of networks were done for each image size. As I read (p.16) "980, 1135, and 1032 points for validation" were used. The authors do not mention if the validation was in-silico only or both in-silico and in real experiments. "

was ignored. I think it is important to present information on real chemical computation because such computation includes errors neglected in the model. For example, the image size 28x28 requires the mixing of almost 800 samples representing different pixels.

Does the volume of a hole in the plate allow for it, or a separate tank is used?

Can the pipetting errors significantly change the result?

It would be interesting if the authors could show an image for which the neural network result differs from the classification with the acid-base system.

The authors should describe more clearly the considered neuron networks. My requests:

"What were the weights of connections?

What were the activation functions of the output neurons?"

have not been taken into account.

I expected the authors could define a matrix with weights corresponding to different classifiers.

The Supplementary Information shows only the weights for the classifier of digits 0 and 1 for 16x16 image in the form of a row of signs.

I am also missing the explicit form of the activation function, for example, for the network presented in Fig.5.

The authors should also optimize the set of weights for a binary network using the information in (Acid, Base) pairs and the majority rule. Would these weights and the network accuracy differ from the networks optimized with a softmax activation function?

The movie attached to the manuscript is a nice illustration of the fact that diffusive mixing is slow and ineffective in small droplets but is completely irrelevant for image recognition. Could the authors include a movie showing the classification of 8x8 image with their device?

And minor mistakes:

p. 2 There is something wrong with Eqs.(4) and (5). What are the values of Acid, Base, x , and y . Would it be simpler to say that a given output is obtained if $n > m$ or $n < m$?

p.3 Symbol x has a double meaning.

I do not recommend the publication of the present form of the manuscript in Nature Communications.

Reviewer #3 (Remarks to the Author):

Sorry for the delay, but I do not work at the University of XYZ anymore, and I do not monitor that e-mail account. I am pretty sure that e-mail account is going to be disabled at the end of this month. Based on this, I think it is better if you find another referee. Otherwise I delegate my review on the other referee. I have to say, so far I agree with everything the other referee said.

I feel like I have iterated this manuscript too many times and it has taken too much of my time. I do not have the time to fully review it again. I thought my last question was quite specific, and the answer from the authors has not convinced me. But as said, I just don't have the time to invest into it a few hours.

I think the fairer thing for the authors would be to either find another fresh referee with the time to review it, or to just rely on the other referee.

Dear Reviewers,

We greatly appreciate your constructive feedback and insights. Accordingly, we have updated the manuscript to clarify the ambiguities highlighted by the reviewers.

The main updates to this version of the manuscript are:

- 1) We provided more details about the experimental setup in addition to the supplementary demonstration video.
- 2) We re-emphasized the usage of Softmax as an activation function.
- 3) We simplified equations 4 & 5 as suggested by the reviewer.
- 4) We updated the figures and table to follow the formatting instructions and editor's guidelines.

Please find our responses below; we have also tried to quote the updated text directly when feasible. Our updates have the following format:

Black: Reviewer's comments.

Blue: Our response to the comment.

Italic: The updated text from the revised manuscript.

Bold: The updated/revised section in the manuscript.

Red: The modified text in the manuscript.

Reviewer #1

- In my opinion, the described chemical classifier can be interesting to a wide audience if it is based on a real working setup. I presume most of the results included in the present version of the manuscript were obtained using numerical simulations of the dual rail processes. My request for precise information on how many experimental tests were made:

"The authors should specify how many experimental tests of networks were done for each image size. As I read (p.16) "980, 1135, and 1032 points for validation" were used. The authors do not mention if the validation was in-silico only or both in-silico and in real experiments. " was ignored. I think it is important to present information on real chemical computation because such computation includes errors neglected in the model. For example, the image size 28x28 requires the mixing of almost 800 samples representing different pixels.

We apologize it wasn't clear in the previous revision. We mentioned that we ran the experimental validation on ten different samples, and we added the breakdown to the manuscript as

- We selected ten random images from the validation dataset for the real experimental evaluation by the Echo liquid handler as follows: one 8x8 binary image (for the 2-class network), one 8x8 binary image (for the 3-class network), one 12x12 binary image (for the 2-class network), one 12x12 binary image (for the 3-class network), and two 16x16 binary images (for the 2-class network) two 16x16 3-bit images (for the 2-class network). **[Page 14]**
- Does the volume of a hole in the plate allow for it, or a separate tank is used?
Thank you for your question; the plate constraints depend on the machine handling. In our case, for Echo 550, we made sure that the volumes we are using do not violate the recommended volume per well for the given plate type. For example, these are the plate constraints for our device:

Plate Type	Minimum Volume per Well	Maximum Volume per Well
384PP	10uL	60uL
384LDV	4uL	12uL
1536LDV	1uL	4uL

In all our reactions (including the final pooling step), the volume per well was within the recommended volume range for the plates; no separate tanks were used. We have updated the manuscript to include the volume limitations of each plate. **[Page 13]**

- Can the pipetting errors significantly change the result?

Thank you for the question; manual pipetting is only used for providing the initial stocks of acids and bases to the liquid handler, and for this step, the volume per well has no impact on the results as long as there is enough volume for the robot to draw acid/base from to encode the computation. After this, the robot handles all the transfers. Additionally, our reactions results are based only on whether the final outputs are acids or bases, so there is large tolerance for accumulated errors since it needs a significant change in the concentration to deviate the output from acid to base or vice versa, especially for applications such as neural networks that are tolerant for errors by nature. So even if the reactions are carried out manually, they should still have a large tolerance for pipetting errors. In order to notice an experimental error that deviates from the pH simulation, the input image, and the discretization level need to be increased significantly such that the Echo's error rate (<10%) or the accuracy of the used concentration can have a significant effect on the final output. However, reaching these high scales would face other obstacles, such as the size of plates that can handle large-scale images. We added a more extensive explanation for the error calculations in the revised manuscripts. **[Page 9-10]**

- It would be interesting if the authors could show an image for which the neural network result differs from the classification with the acid-base system.

We have provided an example in the supplementary material as requested.

[Supplementary Figure 4]

- The authors should describe more clearly the considered neuron networks. My requests: "What were the weights of connections? What were the activation functions of the output neurons?" have not been taken into account.

I expected the authors could define a matrix with weights corresponding to different classifiers.

The Supplementary Information shows only the weights for the classifier of digits 0 and 1 for 16x16 image in the form of a row of signs.

I am also missing the explicit form of the activation function, for example, for the network presented in Fig.5.

Thank you for the feedback; all the weights were provided in the source code attached to the manuscript; additionally, we have added all the weights to the supplementary material as requested.

The activation function used for all our computations was the standard softmax function commonly used across machine learning applications (we mentioned that "We trained a binary neural network consisting of two fully-connected neurons with the Softmax activation"); we have added more references to the activation function earlier in the manuscript.

- We used Softmax [15] as an activation function for all our models. **[Page 7]**

We have also clarified the explicit form in **Figure 5**.

Since we are using binary neural networks conforming with our system, all the weights are either 1 or -1 (represented by + or - signs in the supplementary figure).

We have provided all the weights as tables in the supplementary material and attached with the code. **[Supplementary Table 3-42]**

- The authors should also optimize the set of weights for a binary network using the information in (Acid, Base) pairs and the majority rule. Would these weights and the network accuracy differ from the networks optimized with a softmax activation function?

Thank you for the comment. Since we are using binary neural networks, the computation done by the digital network is identical to the computation done in acids and bases. The softmax generates a higher probability/score for the predicted label, which is the same behavior for the acid-base network that generates higher concentration solutions for the predicted label. Hence, the acid-base model is identical to the *in-silico* model in terms of relative scores given for each label. The reduction in accuracy shown in the tables arises from relying on the pH color indicator for reading the outputs. We rely on the color to

determine the state of each neuron [(Acid, Base) = +1, (Base, Acid) = -1]; which is sufficient when the *in-silico* model produces scores that have a positive sign for the correct label and a negative sign for the wrong label. However, for a few samples, the *in-silico* model and the acid-base model will generate matching results, except that all outputs are either (Acid, Base) or (Base, Acid). However, when we measure the actual pH levels, the output with the highest concentration always matches the prediction label from the *in-silico* model. We have included this explanation in the revised manuscript:

- *The minor reduction in accuracy from the acid-base model compared to the in-silico model arises from relying on the pH color indicator for reading the outputs. We rely on the color to determine the state of each neuron where (Acid, Base) = +1 = On and (Base, Acid) = -1 = Off; which is sufficient when the in-silico model produces scores that have a positive sign for the correct label and a negative sign for the wrong label. However, for a few samples, the in-silico model and the acid-base model will generate matching results, except that all outputs are either (Acid, Base) or (Base, Acid). However, when we measure the actual pH levels, the output with the highest concentration always matches the prediction label from the in-silico model. Supplementary Figure 4 shows an 8x8 image example that cannot be classified correctly with the acid-base network; the pixel values for the image are shown in Supplementary Table 1. [Page 9-10]*

- The movie attached to the manuscript is a nice illustration of the fact that diffusive mixing is slow and ineffective in small droplets but is completely irrelevant for image recognition. Could the authors include a movie showing the classification of 8x8 image with their device?

Thank you for the suggestion; we provide a demonstration movie from the Echo for the 8x8 image classification in the supplementary material.

- And minor mistakes:
p. 2 There is something wrong with Eqs.(4) and (5). What are the values of Acid, Base, x, and y. Would it be simpler to say that a given output is obtained if $n > m$ or $n < m$?

Thank you for the remark. We re-wrote this section using one clear equation:

- *Let x_i denote the type of droplet i (e.g., acid or base). If we consider a mixture of an odd number of n droplets with equal volume and concentrations, with m of these droplets being acids, then:*

$$\text{Majority}(x_1, x_2, \dots, x_n) = \begin{cases} \text{Acid} (= +1) & \text{if } 2m > n \\ \text{Base} (= -1) & \text{if } n > 2m, \end{cases}$$

which is analogous to the majority function in Boolean logic. [Page 2-3]

p.3 Symbol x has a double meaning.

Thank you for the note, in our dual-rail encoding, the concentration of the two sides of the rail should match, i.e., the concentration of the [OH-] should match the concentration of the [H3O+] to utilize the complementary relation used throughout our equations; hence, they are both denoted as the same symbol x . We have added the corresponding, pH equations to the manuscript to make it clearer:

- *An encoded value will be represented by a concentration of [H3O+]= x ions ($\text{pH} = -\log(x)$) in one rail and a concentration of [OH-]= x ions ($\text{pH} = 14.0 - \log(x)$) in the complementary rail. **[Page 3]***